# Mediating trade-off between activity and selectivity in alkynes semi-hydrogenation via a hydrophilic polar layer

Jinqi Xiong[1], Shanjun Mao [1] ✉, Qian Luo[1], Honghui Ning[1], Bing Lu[1], Yanling Liu[1] & Yong Wang [1] ✉

As a crucial industrial process for the production of bulk and fine chemicals, semi-hydrogenation of alkynes faces the trade-off between activity and selectivity due to undesirable over-hydrogenation. By breaking the energy linear scaling relationships, we report an efficient additive-free $WO_3$-based single-atom Pd catalytic system with a vertical size effect of hydrogen spillover. Hydrogen spillover induced hydrophilic polar layer (HPL) with limited thickness on $WO_3$-based support exhibits unconventional size effect to Pd site, in which over-hydrogenation is greatly suppressed on $Pd_1$ site due to the polar repulsive interaction between HPL and nonpolar C=C bonds, whereas this is invalid for Pd nanoparticles with higher altitudes. By further enhancing the HPL through Mo doping, activated $Pd_1/MoWO_3$ achieves recorded performance of 98.4% selectivity and 10200 $h^{-1}$ activity for semi-hydrogenation of 2-methyl-3-butyn-2-ol, 26-fold increase in activity of Lindlar catalyst. This observed vertical size effect of hydrogen spillover offers broad potential in catalytic performance regulation.

The semi-hydrogenation of alkynes to alkenes is a crucial and versatile chemical reaction with numerous industrial and synthetic applications, ranging from bulk to fine chemicals[1,2]. The most widely used catalyst for this reaction is the supported Pd catalyst, which often displays limited selectivity due to side reactions like oligomerization, isomerization, and over-hydrogenation, caused by strong adsorption of alkenes on the Pd site[3,4]. The strong adsorption of alkyne can also result in reactant-induced poisoning of the catalyst[5–8]. The Lindlar catalyst, an industrial benchmark, addresses these issues by poisoning $Pd/CaCO_3$ with lead and quinoline, thus enhancing selectivity. However, its low activity and toxic additives limit its application in modern industry.

Inspired by the Lindlar catalyst, recent studies have shown that various ligands (e.g., polyamines[5,9], thiolate[10,11], silane[12]) can create a metal-organic interface to adjust reactant adsorption behaviors. One example is the NanoSelect™ catalysts developed by BASF, which use hexadecyl-2-hydroxyethyl-dimethyl ammonium dihydrogen phosphate

(HHDMA) as a ligand, achieving a selectivity (mostly > 95%) for semi-hydrogenation of alkynes without toxic additives[13–15]. However, the ligand's strong adsorption on Pd restrains its hydrogenation activity. Other approaches include constructing a strong metal-support interaction[16–18], alloying Pd with inert components (i.e., Bi[19], Ga[20], In[21,22], Zn[23,24], Ag[25], Au[26], Cu[27,28], and even subsurface carbon[29]) or downsizing Pd to single atoms[30–32] to regulate the electronic structure as $d$-band center and weaken C=C bond adsorption. Unfortunately, these methods often compromise activity to achieve higher selectivity. The fundamental cause of the activity-selectivity trade-off lies in the overlapping hydrogenation kinetic of C≡C and C=C bonds, leading to the so-called energy linear scaling relationships (LSRs)[33–36]. The key to resolving this paradox lies in breaking the LSRs, which remains an underexplored area of research.

Utilizing spillover hydrogen serves as a promising alternative since hydrogen activation and addition processes are separated and occur on the main active site of metal and the secondary active site of

[1]Advanced Materials and Catalysis Group, Center of Chemistry for Frontier Technologies, State Key Laboratory of Clean Energy Utilization, Institute of Catalysis, Department of Chemistry, Zhejiang University, Hangzhou 310058, P. R. China. ✉e-mail: maoshanjun@zju.edu.cn; chemwy@zju.edu.cn

support, respectively[37–42]. For instance, dissociated hydrogen that spilled from encapsulated noble metal nanoparticles (NPs) to the support exhibits activity for the hydrogenation of alkynes while remaining inert for the hydrogenation of alkenes[37]. However, this method suffers from very limited hydrogenation activity compared to noble metal sites like Pd. Considering the non-polar properties of C≡C and C=C, their adsorption strength on the active site can be directly influenced by the polarity of the surrounding micro-environment. Thus, it is feasible to leverage the mutual repulsion between C≡C/C=C and the active site by increasing the surface polarity to weaken or even inhibit the adsorption of C=C on the noble metal site from a thermodynamic perspective. One desirable option is to construct a hydrophilic polar layer (HPL) composed of a hydroxyl array through hydrogen spillover on reducible metal oxides. This strategy offers the advantage of not significantly altering the intrinsic activity of the metal active sites while improving selectivity, as their electronic structures are not directly regulated. Consequently, it breaks the LSRs, as depicted in Fig. 1a. Furthermore, spillover hydrogen on the support can provide an additional activity pathway. However, to the best of our knowledge, no published reports have discussed this approach. It is worth noting that the thickness of the HPL is limited to the height of the hydroxyl groups. Therefore, this strategy is effective only when the vertical size of the metal site matches the thickness of the HPL and the selectivity regulation by the HPL on the semi-hydrogenation of C≡C should demonstrate a vertical size effect of the metal active sites, as illustrated in Fig. 1b.

In this work, a series of supported Pd catalysts with Pd size ranging from single atom to large particles were synthesized to explore the vertical size effect of hydrogen spillover on the selectivity regulation for semi-hydrogenation of alkynes. As expected, Pd catalysts with various particle sizes on hydrogen spillover-prone supports exhibited a significant enhancement in alkyne semi-hydrogenation activity. However, only the single-atom Pd catalysts, such as $Pd_1/WO_3$, demonstrated a remarkable improvement in alkene selectivity following $H_2$ pretreatment, where the vertical size effect of hydrogen spillover was observed. We discovered that hydrogen spillover led to the formation of a hydrophilic polar layer (HPL) comprising a hydroxyl array on the $WO_3$ surface. This HPL influenced the semi-hydrogenation reaction and exhibited sensitivity to metal particle size. In the case of Pd NPs, the limited thickness of the HPL prevented it from affecting the adsorption of C=C bond on the metal surface, resulting in no improvement in selectivity (Fig. 1b). However, the presence of spillover hydrogen at the metal-support interface enhances the reaction activity. On the other hand, for single-atom Pd sites, the thickness of the HPL ensured its repulsive effect on both non-polar C≡C and C=C bonds adsorbed on the single-atom Pd surface. Moreover, the addition of Mo facilitated hydrogen spillover and enhanced the HPL, further improving the catalytic performance. This vertical size effect of the HPL could be extended to other selective hydrogenation reactions, such as p-chloronitrobenzene and nitrostyrene, and represents a novel approach to address the trade-off between activity and selectivity using hydrogen spillover.

## Results and discussion
### Evidence and effect of hydrogen spillover
To investigate the impact of hydrogen spillover on the semi-hydrogenation of alkynes, we conducted comparative tests using various supports, including reducible ($WO_3$, $TiO_2$, $CeO_2$) and irreducible ($Al_2O_3$, $SiO_2$, MgO) supports loaded with -0.3 wt% uniform Pd nanoparticles (NPs) with an average size of 2.8 nm (Supplementary Figs. 1–3)[43]. Detailed characterizations are provided in Supplementary Fig. 4–5. The semi-hydrogenation of 2-methyl-3-butyn-2-ol (MBY), a significant process for producing vitamins and spices, was conducted as a probe reaction to evaluate the performance of those Pd-based catalysts before and after hydrogen pretreatment (introducing hydrogen spillover). The results presented in Fig. 2a and b demonstrated that, after hydrogen pretreatment, Pd catalysts supported on reducible supports exhibited a notable improvement in activity, while those with unreducible supports displayed negligible activity enhancement (Supplementary Fig. 6). No detectable variation in selectivity was found among these Pd NPs catalysts, whether activated or not (Supplementary Fig. 7). This difference in activity strongly suggests that hydrogen spillover plays a significant role in the hydrogenation process[44]. Based on the finding that $Pd/WO_3$ exhibited the most significant catalytic activity increment (4-fold) and relatively great selectivity, we selected $WO_3$ as the support to investigate the underlying mechanism for the observed activity improvement.

Aberration-corrected HAADF-STEM images indicated that the Pd species existed in the form of particles, and no small clusters or single atoms were observed (Supplementary Fig. 8–10). After hydrogen pretreatment, amorphous $H_xWO_3$ was observed on the $Pd/WO_3$ surface, with the lattice fringe of $WO_3$ becoming blurred at the edge (Fig. 2c)[45]. TEM and XRD pattern did not reveal any discernible differences in the bulk phase, suggesting the $H_xWO_3$ was mainly formed on the surface (Supplementary Fig. 11, 12). In situ $H_2$ infrared spectroscopy showed that the peaks for stretching vibration of W-OH at 3290 cm$^{-1}$ emerged and enhanced with increasing hydrogen exposure time (Fig. 2d and Supplementary Fig. 13), while switching the atmosphere from $H_2$ to $D_2$, the signals for the new O-D entities on the support could be spotted easily (Supplementary Fig. 14), which demonstrated the occurrence of hydrogen spillover[46,47]. Similar results were confirmed by the proportion rise in the hydroxyl groups at 531.5 eV in XPS spectra of O 1$s$ from 21% to 35% (Fig. 2e), the positive shift of bridging hydroxyl groups at 4.54 ppm, and the occurrence of terminal hydroxyl groups at 1.08 ppm in $^1$H NMR (Fig. 2f)[48–51]. These results indicate that HPL gradually formed during the hydrogen pretreatment process, which is closely related to the improved catalytic performance of $Pd/WO_3$.

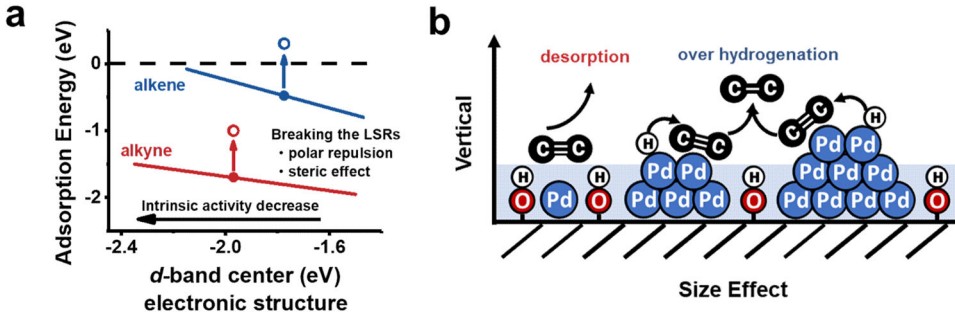

**Fig. 1 | Schematic illustration of breaking the linear scale relationships and size effect. a** Adsorption energy regulation to break the energy linear scale relationships. **b** Vertical size effect of the hydrophilic polar layer.

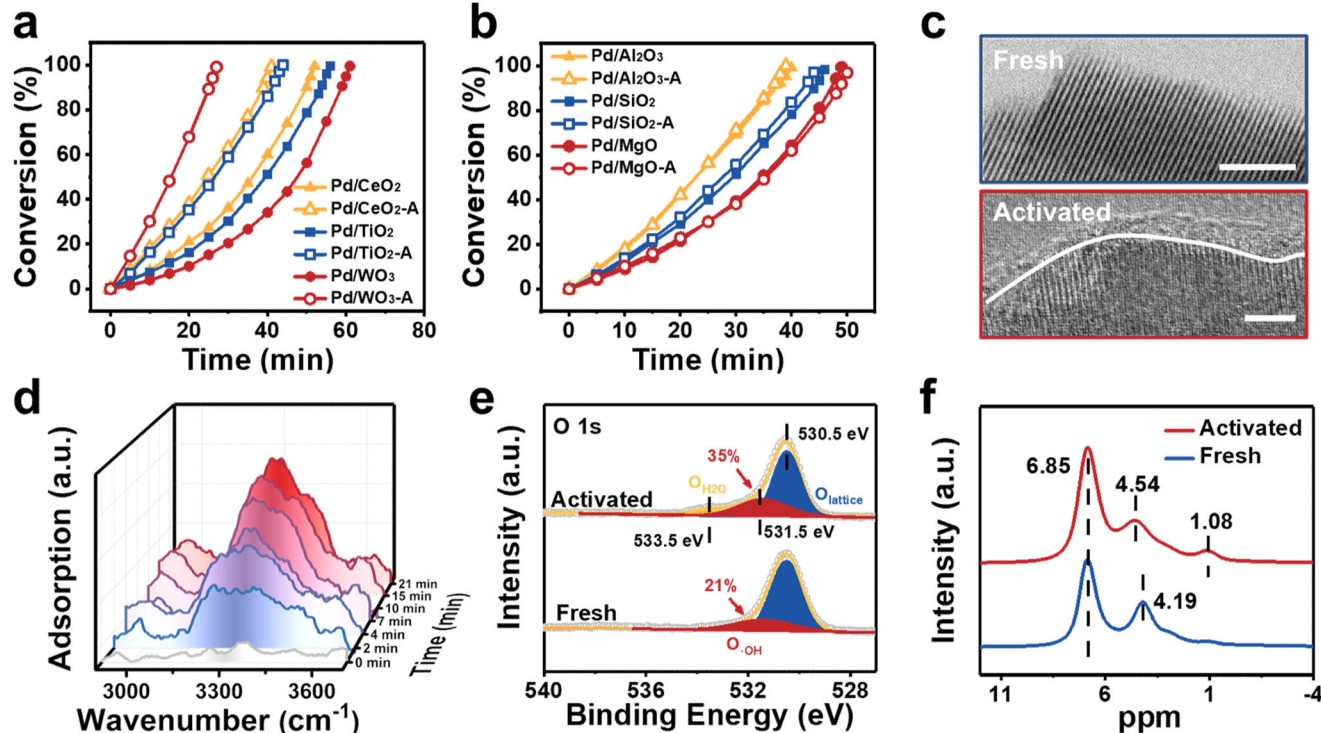

**Fig. 2 | Comparison of catalytic performance and structural characterizations of Pd/WO$_3$.** Comparison for the catalytic activity of the MBY hydrogenation with and without hydrogen pretreatment over reducible supports (**a**) and irreducible supports (**b**). Reaction condition: 10 ml of ethanol, 40 °C, 1 bar H$_2$, and 1000 rpm. 1 mmol substrate, 0.028 mol% Pd. **c** HRTEM of the fresh Pd/WO$_3$ and activated Pd/WO$_3$. Scale bar: 2 nm. **d** Evolution of in situ FT-IR spectra of H$_2$ adsorbed on Pd/WO$_3$ in a flow of H$_2$. High-resolution XPS spectra of O 1$s$ specimens (**e**) and 400 MHz $^1$H MAS solid-state NMR spectra (**f**) for fresh Pd/WO$_3$ and activated Pd/WO$_3$.

## Investigation in activity contributions

In order to explore the connection between hydrogenation kinetics and the extent of hydrogen spillover, we conducted tests on the semi-hydrogenation activity of Pd/WO$_3$ under different hydrogen pretreatment durations. As shown in Fig. 3a and Supplementary Fig. 15, the catalytic activity increased gradually with longer activation time, ultimately reaching a steady state with activity 4 times higher than that of the fresh catalyst[52]. Successive batch experiments exhibited that the activity significantly improved in the second run and remained stable thereafter, indicating that the Pd/WO$_3$ could also be in-situ activated during the hydrogenation without hydrogen pretreatment (Supplementary Fig. 16). Furthermore, the reaction rate of MBY was detected to increase with time on fresh Pd/WO$_3$, with a reaction order of −0.95 (Fig. 3b). The negative reaction order suggests that the excessive adsorption of alkynes induces a competitive behavior, blocking Pd active sites and hindering the activation of H$_2$[23]. As the pretreatment time was extended, the reaction order of MBY gradually reduced to −0.25, suggesting that the HPL formed by hydrogen spillover could efficiently weaken the competitive inhibition of MBY without directly modifying the electronic structure of the active Pd sites (Supplementary Fig. 17), thereby enhancing the activity. In addition, the kinetic isotope effect (KIE) was explored on activated Pd/WO$_3$ (Pd/WO$_3$-H) using D$_2$ instead of H$_2$ (Supplementary Fig. 18). The calculated KIE value ($k_H/k_D$ = 1.64) indicated that rate-limiting step is MBY hydrogenation rather than H$_2$ activation, further proving that the HPL effectively reduced the poisoning effect of MBY.

We further prepared Pd/WO$_3$ catalysts with highly dispersed Pd atoms as well as Pd NPs of 4.6 and 7.8 nm (Supplementary Figs. 19–22) to investigate the effectiveness of hydrogen-pretreated activation. The atomic dispersion of Pd in Pd$_1$/WO$_3$ was confirmed through extended X-ray absorption fine structure (EXAFS) analysis, which showed a distinct peak of Pd–O contribution at 1.52 Å and the absence of Pd–Pd coordination at 2.55 Å (Fig. 3c). Additional characterization results are presented in Supplementary Figs. 23–26 and Supplementary Table 2. All Pd/WO$_3$ catalysts with different Pd ensemble sizes exhibited enhanced activity after hydrogen pretreatment, with smaller Pd ensembles displaying a greater increase in activity (Fig. 3d and Supplementary Fig. 27). Among these, Pd$_1$/WO$_3$-H demonstrated the highest performance with a 6-fold increase in activity.

To gain a deeper understanding of the intrinsic chemical mechanism behind the improvement of catalytic activity of Pd/WO$_3$ via hydrogen spillover, we varied the particle size (single atom, 2.8 nm, 4.6 nm, 7.8 nm) and studied the effect of catalyst loading (0.05–0.30 wt%) on activity (Fig. 3e and Supplementary Fig. 28). The figure revealed a good linear relationship between catalyst loading and conversion rate, but all the correlation lines do not pass through the origin, indicating that the support itself exhibits some activity[31,39]. Similar conclusions were drawn in the selective poisoning experiments that used CO and Li$_2$SO$_4$ as targeted inhibitors to poison the Pd and support sites of Pd$_{NP}$/WO$_3$ (WO$_3$ supported Pd nanoparticle catalyst), respectively (Supplementary Figs. 29–31)[53,54]. Additionally, we investigated the reaction rates normalized by Pd loading to distinguish the contribution of Pd sites (Fig. 3f). As the Pd loading increased, the normalized activity tended to remain constant, which indicated that the contribution of support sites weighed more with the lower Pd loading (Supplementary Fig. 32). Based on these findings, we proposed an underlying mechanism for the improved activity by hydrogen spillover on Pd/WO$_3$ (Fig. 3g). Pd sites played a primary role in H$_2$ dissociation and substrate hydrogenation, while WO$_3$ functioned as a medium to transport and store H atoms, forming H$_x$WO$_3$ as a dynamic "hydrogen pool" to participate in the hydrogenation. This dynamic process simultaneously avoided the competitive adsorption of substrates and H$_2$ and accelerated the hydrogenation rate.

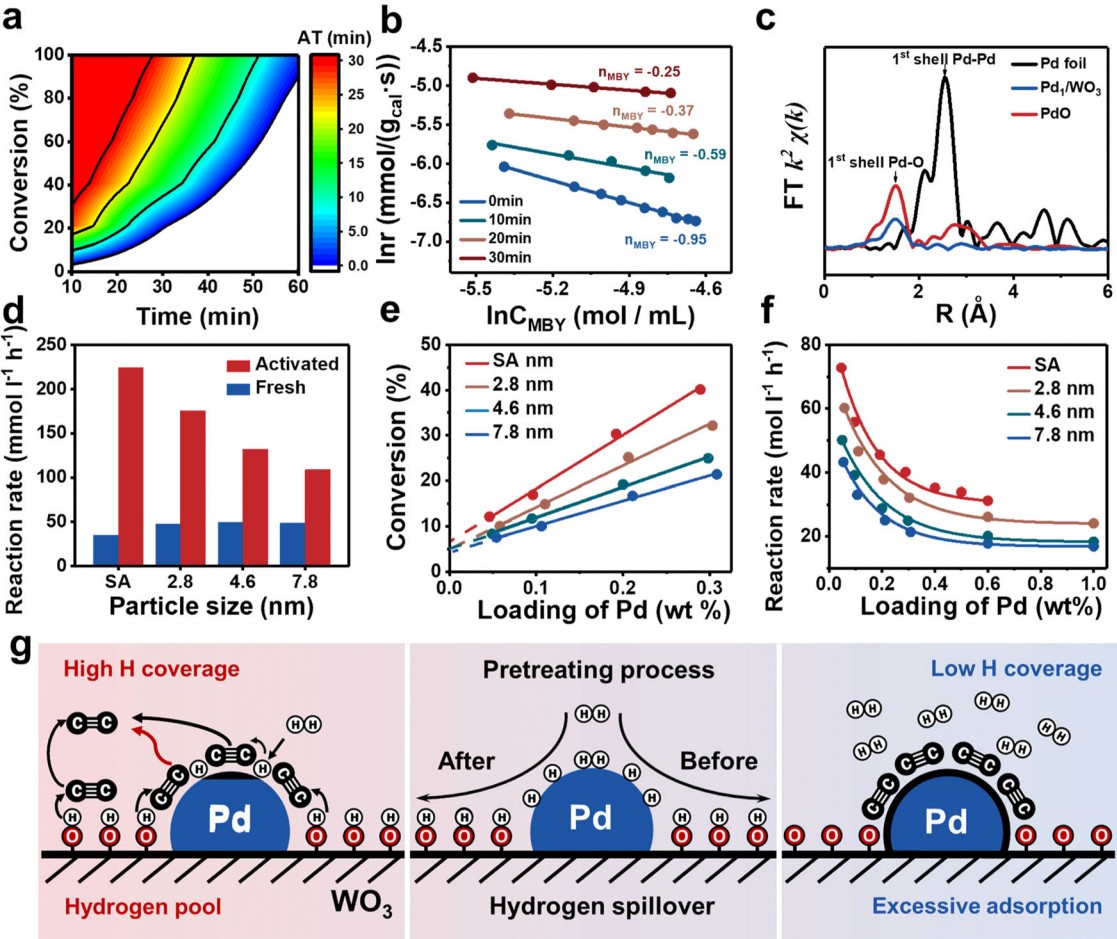

**Fig. 3 | Root of the activity improvement in Pd/WO₃. a** The catalytic performance over Pd/WO₃ after pre-activation in H₂ for different period (AT: activated time). **b** Reaction orders of MBY calculated over Pd/WO₃ related to the pretreated time. **c** Fourier transform (FT) EXAFS spectra of Pd₁/WO₃, PdO and Pd foil at the Pd K edge. χ, the relative modulation of EXAFS; k, the photoelectron wave number relative to the absorption edge energy; R, radial distance. **d** Reaction rate for the hydrogenation of MBY over fresh and activated Pd/WO₃ loaded different size-controlled Pd NPs and single atoms. **e** The conversion of MBY within 10 min over H₂ pretreated Pd/WO₃ of different loadings of Pd. **f** Reaction rate normalized by Pd loading over H₂ pretreated Pd/WO₃ of different loadings of Pd. Reaction condition: 10 ml of ethanol, 40 °C, 1 bar H₂, and 1000 rpm. 1 mmol substrate, 10 mg catalyst. **g** Schematic depiction of the contribution of overflowed H to hydrogenation through "hydrogen pool".

## Mechanism of HPL in selectivity improvement

Besides its activity, the ability to control selectivity towards alkenes is an even more desirable feature. Surprisingly, the Pd₁/WO₃ exhibited significantly different selectivity compared to Pd$_{NP}$/WO₃, as shown in Fig. 4a and Supplementary Fig. 33a. The selectivity of MBE on fresh Pd₁/WO₃ declined quickly to 88.9%, while it maintained 95.4% on Pd₁/WO₃-H. Moreover, negligible improvement in selectivity was observed for Pd$_{NP}$/WO₃ with higher altitudes. Subsequently, Pd₁/WO₃ with varying hydrogen pretreatment times were evaluated and both activity and selectivity gradually increased when extending the hydrogen pre-treatment time (Fig. 4b). After full conversion of MBY, MBE hydrogenation was also significantly inhibited on Pd₁/WO₃-H, resulting in only a slight decrease in selectivity from 95.4% to 84% over 2 h, which was much higher than that of fresh Pd₁/WO₃ (from 88.5% to 55.5%). As expected, the yield of MBY decreased sharply from 84% to 24% for the Pd NPs counterparts (Supplementary Figs. 33b and 34).

The presented results highlight the significant role of the HPL formed on Pd₁/WO₃-H in suppressing over-hydrogenation by blocking the Pd site, with this blocking effect being size-sensitive. On one hand, the HPL provides proton shielding through polar incompatibility, preventing the adsorption of non-polar molecules or functional groups like alkynes and alkenes on the catalytic surface. On the other hand, the bond length of W-OH (2.35 Å) suggests a size effect in the

vertical surface direction for proton shielding. In the case of Pd₁/WO₃-H, the Pd site was shielded by the HPL, as the atomic radius of Pd (1.37 Å) is shorter than the bond length of W-OH. Conversely, the surface sites of Pd particles in Pd$_{NP}$/WO₃ remain accessible to alkenes for hydrogenation, leading negligible selectivity improvement (Fig. 1b). To support our hypothesis, a series of verified experiments were further conducted.

Contact angle measurements were performed to investigate the change in hydrophilicity of the prepared catalysts[55,56]. The water contact angle (WCA) of Pd₁/WO₃ decreased from 42°4′ to 12°24′ as the hydrogen pretreatment time prolonged (Fig. 4c and Supplementary Fig. 35), indicating that the hydrogen treatment made the catalyst surface more hydrophilic and oleophobic[42,57]. For alkenes, the polar repulsive interaction and the steric effect between hydroxyl groups and alkenes were beneficial to the desorption. An increase in selectivity to MBE with lower WCA was expected (Supplementary Fig. 36). For alkynes, the repulsion reduced the poisoning effect of C≡C, promoting the H₂ dissociation and alkyne hydrogenation. Temperature-programmed desorption (TPD) of C₂H₂ and C₂H₄ was conducted to elucidate the variations in substrate adsorption on Pd₁/WO₃ and Pd₁/WO₃-H. As shown in Fig. 4d, e, C₂H₂ and C₂H₄ adsorbed on fresh Pd₁/WO₃ desorbed at 388 K and 373 K, respectively. The higher desorption temperature and peak intensity of C₂H₂ indicated its stronger

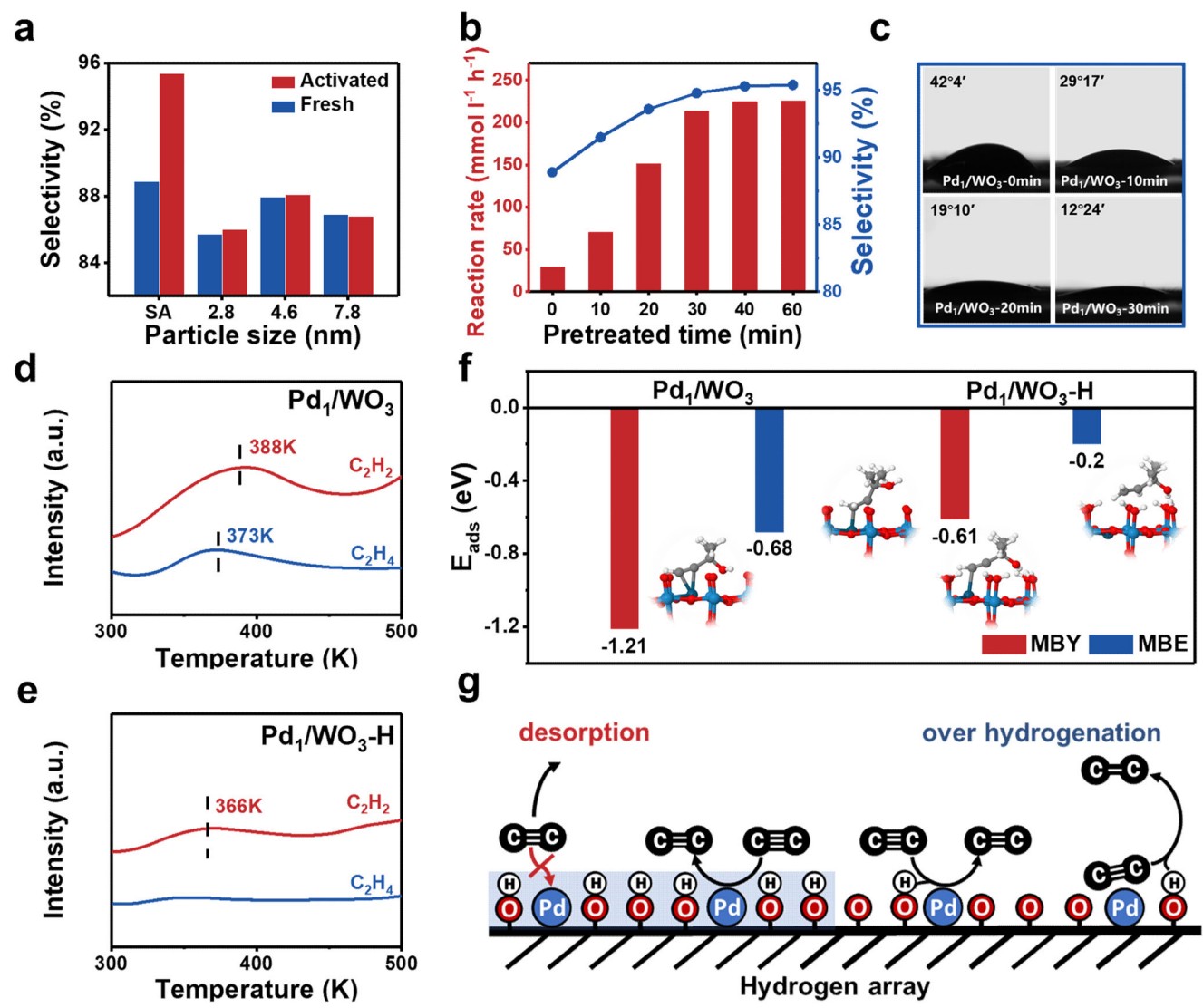

**Fig. 4 | Mechanism behind the unconventional selectivity promotion in Pd₁/WO₃. a** The selectivity to MBE at full MBY conversion over fresh and activated Pd₁/WO₃ loaded different size-controlled Pd NPs and single atoms. **b** Reaction rate and selectivity over Pd/WO₃ with different pretreated time of hydrogen. **c** Evolution of the water contact angle over Pd₁/WO₃ after hydrogen pretreatment. Temperature-programmed desorption experiments of C₂H₂ and C₂H₄ over Pd₁/WO₃ (**d**) and Pd₁/WO₃-H (**e**). **f** DFT calculations. Configurations and the calculated E_ads values of MBY and MBE adsorption on Pd₁/WO₃ and Pd₁/WO₃-H. Color code – dark blue: Pd, light blue: W, grey: C, white: H. **g** Schematic depiction of influence of the HPL on the single atoms.

adsorption[58,59]. After hydrogen pretreatment, C₂H₄ adsorption became negligible, and the peak of C₂H₂ desorption shifted to lower temperature (366 K), indicating that HPL could effectively reduce both the adsorption strength of alkyne and alkene through the polar repulsion. Density functional theory (DFT) calculation in Fig. 4f further confirmed that the adsorption energy (E_ads) of MBE significantly decreased from −0.68 eV to −0.2 eV after HPL formation, which was weak enough to prevent the re-chemisorption and further hydrogenation of MBE. Simultaneously, the E_ads of MBY decreased from −1.21 eV to −0.61 eV, consistent with the weakened poisoning effect. The E_ads ratio of MBY to MBE on Pd₁/WO₃-H was enlarged, also indicating increased MBE selectivity. When examining the adsorption configurations, one can assumed the decrease of E_ads for MBY and MBE resulted from the reduced free reaction space above Pd sites.

A plausible reaction mechanism explaining the selectivity promotion on Pd₁/WO₃ due to the HPL is proposed. Upon H₂ pretreatment, the HPL formed through hydrogen spillover on Pd₁/WO₃-H envelops the single Pd site, offering proton shielding to non-polar functional groups through polar incompatibility. This dynamical repulsion of the as-formed alkenes hinders over-hydrogenation (Fig. 4g). However, due to the size limitation of the hydroxyl group array, the proton shielding effect of HPL is restricted to the sub-nanometer scale in the vertical direction. Consequently, some Pd sites remain accessible to alkenes for subsequent hydrogenation on Pd NPs (Fig. 1b). As a result, the unique selectivity promotion phenomenon with hydrogen spillover is observed only in the case of WO₃-supported single-atom Pd catalysts in alkyne semi-hydrogenation. This strategy of constructing HPL around Pd single atom effectively regulates the adsorption of alkenes and alkynes without modifying the electronic structure of the active sites, successfully breaking the linear scaling relationships (LSRs) and hence enhancing selectivity without compromising the intrinsic activity of the catalyst. It should be noted that the effect of the HPL is also applicable to other single atom catalysts with reducible supports where hydrogen spillover occurs. For instance, the hydrogen-pretreated Pd₁/TiO₂ displayed increased activity and selectivity in MBY semi-hydrogenation, while Pd₁/Al₂O₃ did not show the same effect (Supplementary Fig. 37).

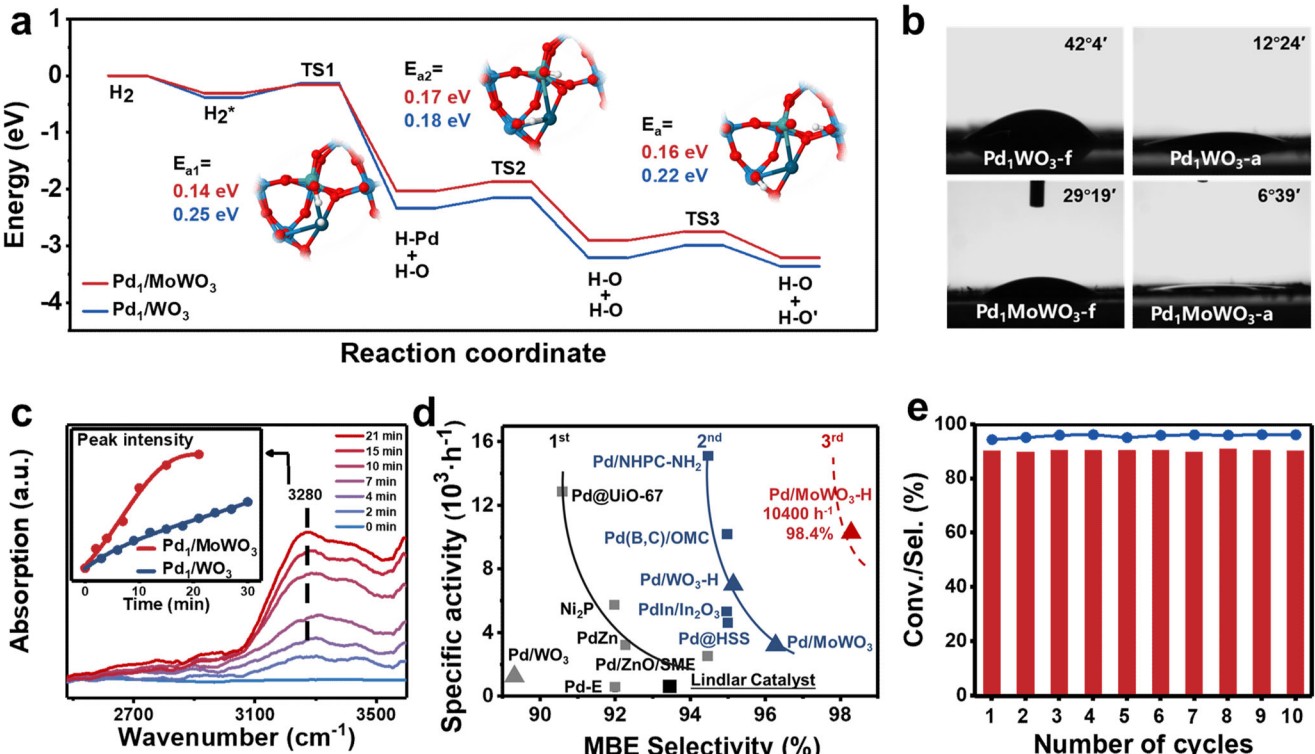

**Fig. 5 | Impact of Mo doping on structural properties and catalytic performance of Pd₁/MoWO₃. a** Structures of the transition states of hydrogen spillover process on Pd₁/MoWO₃ and DFT-calculated reaction energies of the heterolytic H₂ dissociation and subsequent spillover on adjacent (H–O) for Pd₁/WO₃ and Pd₁/MoWO₃ (detailed structures were shown in Supplementary Fig. 38). **b** Contact angle measurements for activated Pd₁/MoWO₃ and Pd₁/WO₃. **c** Evolution of in situ FT-IR spectra of H₂ adsorbed on Pd/MoWO₃ in a flow of H₂ and the absorption intensity at 3280 cm⁻¹ toward time over Pd₁/WO₃ and Pd₁/MoWO₃. **d** Comparison of specific activity and MBE selectivity for fresh Pd₁/WO₃, activated Pd₁/WO₃, fresh Pd₁/MoWO₃, activated Pd₁/MoWO₃ and other catalysts toward semi-hydrogenation of MBY. The specific reaction conditions and conversion of each catalyst can be found in the Supplementary Table 2. **e** Cycling stability test of the selective hydrogenation of MBE over Pd₁/MoWO₃.

## Pd₁/MoWO₃-catalyzed selective hydrogenation

Subsequent efforts were aimed to enhancing HPL in Pd/WO₃ to further improve its catalytic properties. As reported in the previous literature, moderate doping of Mo can reduce the band-gap of WO₃, promoting the proton migration and insertion process[60–62]. DFT calculation results depicted in Fig. 5a revealed that the barrier for H₂ heterolytic cleavage (TS1) was lower on Pd₁/MoWO₃ (0.14 eV) compared to Pd₁/WO₃ (0.25 eV). This indicated that Mo doping could effectively increase H₂ activation ability. The same conclusion could be made for the subsequent spillover hydrogen migration[63].

A trial was then conducted to dope Mo into WO₃ to facilitate hydrogen spillover. Detailed characterizations of Pd₁/MoWO₃ were shown in Supplementary Fig. 39–45. In-situ IR spectroscopy confirmed the acceleration of hydrogen spillover, as the variation rate of hydroxy peak intensity at 3280 cm⁻¹ increased by a factor of 3.6 on Pd₁/MoWO₃ (Fig. 5b and Supplementary Fig. 46). WCA measurements also demonstrated that Pd₁/MoWO₃ was more hydrophilic than Pd₁/WO₃, indicating a denser HPL and stronger mutual repulsion on Pd₁/MoWO₃ (Fig. 5c). DFT calculation in Supplementary Fig. 47 further showed that the adsorption energy of MBE and MBY on both Pd₁/MoWO₃ and Pd₁/MoWO₃-H significantly decreased, which was further proved by the temperature programmed desorption (TPD) of C₂H₂ and C₂H₄ (Supplementary Fig. 48). The adsorption of alkene is dramatically decreased compared to alkyne after the formation of HPL, which greatly improves the selectivity by preventing the over hydrogenation[23,64].

As anticipated, fresh Pd₁/MoWO₃ exhibited 1.5 times higher activity and improved selectivity of MBE, reaching up to 96.1% at >99% conversion, compared to Pd₁/WO₃ (Supplementary Fig. 49).

The induction period was also reduced to 2/3 of the undoped case, indicating a faster hydrogen spillover process (Supplementary Fig. 50). Ultimately, the activated Pd₁/MoWO₃-H catalyst demonstrated an impressive yield of MBE up to 98.4% and a 26-fold activity (10200 h⁻¹ at 298 K) increase compared to commercial Lindlar catalysts for the semi-hydrogenation of MBY (Supplementary Fig. 51). This performance not only ranks as the best result among reported literatures (much better than the reported catalytic performance), but also declare the arrival of the era of third-generation catalyst development for semi-hydrogenation of alkynes (Fig. 5d and Supplementary Table 3).

Pd₁/MoWO₃-H with HPL also exhibited excellent stability with the activity and selectivity remaining unchanged in at least 10 subsequent cycles (Fig. 5e and Supplementary Fig. 52). Additionally, Pd₁/MoWO₃-H demonstrated a wide substrate scope for various alkynes, with the highest selectivity of 99.6% achieved for (Z)-4-octene at full conversion (Fig. 6 and Supplementary Fig. 53). Importantly, a sluggish over-hydrogenation phenomenon was observed after the full conversion of internal alkyne compounds (10a-14a), which is significant for determining the reaction endpoint in industrial applications. To account for industrial reaction conditions, a solvent-free experiment was conducted on Pd₁/MoWO₃, resulting in an impressive > 99% yield of MBE in 14 h (Supplementary Fig. 54). Notably, the proton shielding effect originated from HPL also improved the selectivity for other hydrogenation reactions, such as p-chloronitrobenzene (95.8% to 98.9%) and 3-nitrostyrene (64.9% to 83.2%) (16a–17a).

In summary, we have developed a WO₃-based single-atom Pd catalyst featuring a unique hydrogen spillover size effect, which demonstrates distinguished catalytic performance for additive-free

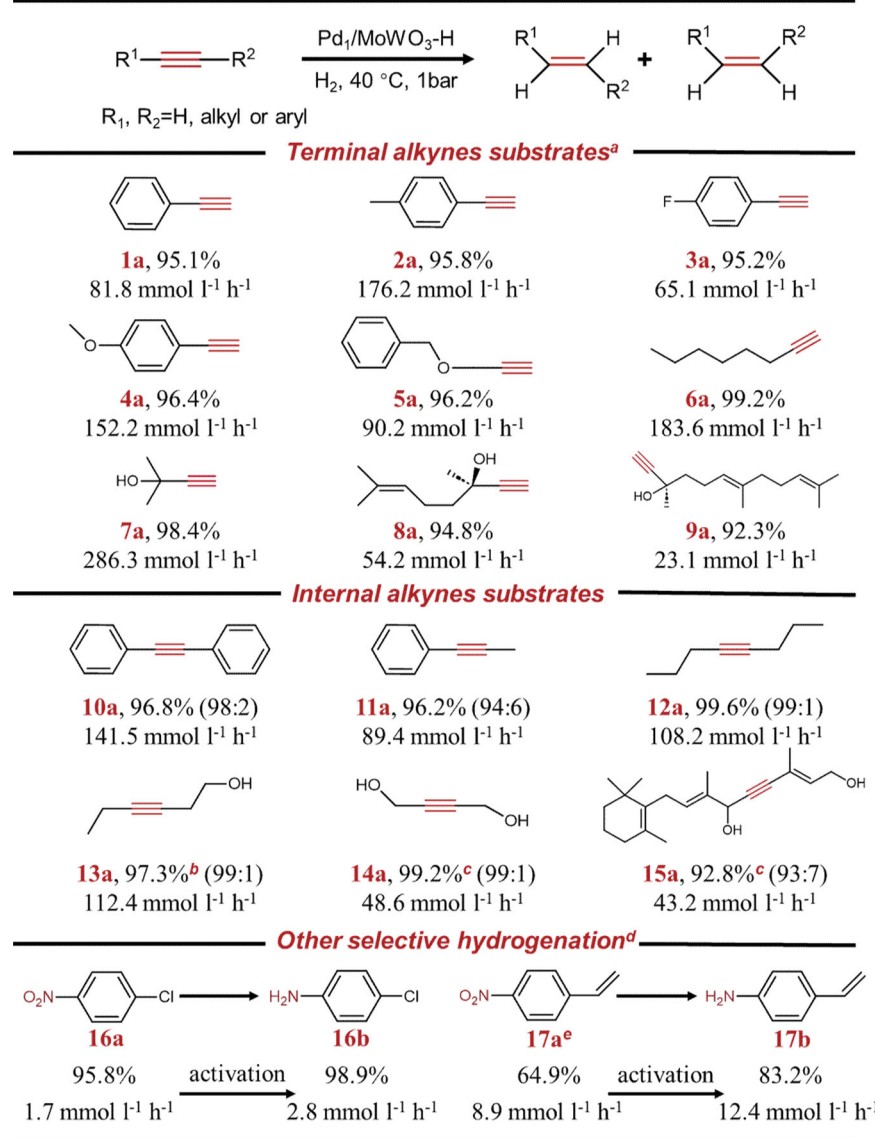

**Fig. 6 | Substrate scope of various alkynes and other substrates over Pd₁/MoWO₃-H.** The red color represents the hydrogenation of C≡C bonds to corresponding C=C, and the ratio within brackets for the internal alkyne substrates indicates the cis/trans ratio of alkene products. ᵃReaction condition: 10 mL of ethanol, 40 °C, 1 bar H₂, and 1000 rpm. 1 mmol substrate, 0.028 mol% Pd. ᵇ60 °C. ᶜ80 °C, 0.084 mol% Pd. ᵈ0.056 mol% Pd, 1 Mpa H₂. ᵉ 0.056 mol% Pt. Detailed characterizations of Pt₁/MoWO₃ are provided in Supplementary Fig. 55-57.

semi-hydrogenation of alkynes. During H₂-pretreatment, a hydrophilic polar layer (HPL) forms on the catalyst surface, serving as a hydrogen reservoir that provides an additional activity channel. Furthermore, HPL imparts strong polarity to the micro-environment surrounding the Pd sites. The polar incompatibility between HPL and reactants results in mutual repulsion, weakening the adsorption of nonpolar C=C and C≡C bonds, which finally suppresses the over-hydrogenation and reduces the poisoning effect of alkynes. Due to the height limitation of hydroxyl group array, a vertical size effect of hydrogen spillover is observed on Pd₁/WO₃ and further enhanced by Mo doping, which exhibits exceptional performance in the selective hydrogenation of MBY and a variety of other substrates, rendering selectivity regulation unavailable for Pd nanoparticle counterparts. The vertical size effect of hydrogen spillover efficiently breaks the linear scaling relationships (LSRs) and is also applicable for other selective hydrogenation reactions, offering more possibilities for selectivity control in challenging reactions, and perhaps more importantly the expansion of the research field on hydrogen spillover.

## Methods

### Chemicals and materials

Oxide supports, including γ-Al₂O₃, SiO₂, MgO, CeO₂, and TiO₂ were purchased from rom Aladdin Industrial Corporation and calcined at 500 °C for 1 h. All chemicals were used as purchased without further purification and were commercially available: These chemicals include citric acid (AR), ethylene glycol (AR), acetone (AR), ammonium meta-tungstate, ((NH₄)₆H₂W₁₂O₄₀· xH₂O, AR), ammonium molybdate ((NH₄)₆Mo₇O₂₄· 4H₂O, AR), polyvinyl pyrrolidone (MW=58000, AR), palladium chloride (PdCl₂, AR), ammonium bicarbonate (NH₄HCO₃, AR), sodium borohydride (NaBH₄, AR), lithium sulfate (Li₂SO₄, AR), 2-methyl-3-butyn-2-ol (MBY, AR) and other alkyne compounds. Lindlar catalyst (5% palladium on calcium carbonate poisoned by lead) was purchased from Aladdin.

### Synthesis of catalysts

The Pd₁/WO₃ catalysts were synthesized via the wetness impregnation method. Specifically, 300 mg of WO₃ was dispersed in 15 ml of deionized

water and stirred for 30 min. Simultaneously, 273 mg of $NH_4HCO_3$ was dissolved in 20 ml of deionized water, followed by sonication for 30 min. The resulting solutions were then combined and stirred for an additional 30 min, resulting in a flocculent blue precipitate. Next, 100 mg $PdCl_2$ was first dissolved in 10 ml hydrochloric acid solution (0.6 M) to obtain $PdCl_2$ solution. 150 μL $PdCl_2$ hydrochloric acid solution (containing 6 mg·ml$^{-1}$ Pd) diluted with 8 ml deionized water was added in the $WO_3$ dispersion and constantly stirred for 1 h. The resulting product was separated via centrifugation and subsequently dried under vacuum at 70 °C. Finally, the catalysts were reduced in $H_2$ (flow rate = 50 ml·min$^{-1}$, heating rate=2 °C·min$^{-1}$) at 150 °C for 1 h, yielding 0.3% $Pd_1/WO_3$. The Pd/$MoWO_3$ catalysts were prepared using the same protocol, with $MoWO_3$ was used as the support material. The loading amount of Pd was regulated by the volume of the precursor solution.

## Characterizations

Scanning electron microscopy (SEM) study was performed on a Hitachi SU8010 microscope. An electron paramagnetic resonance (EPR) experiment was conducted on a Bruker A300-10/12. The high-resolution Transmission Electron Microscope (HRTEM) and EDS mapping was conducted on the JEOL JEM-2100F at an acceleration voltage of 200 kV. Spherical aberration corrected Transmission Electron Microscope (HADDF-STEM) was performed on a FEI Titan G280-200 ChemiSTEM at an acceleration voltage of 200 kV. Power X-ray diffraction (XRD) patterns were performed on a Rigaku Ultima IV operating at 40 kV and 20 mA with Cu Kα radiation. The inductively coupled plasma-optical emission spectrometry (ICP-OES) was performed on a Perkin Elmer Optima OES 8000. Raman spectrum was collected on a LabRam HRUV. The X-ray photoelectron spectra (XPS) was obtained on an ESCALAB MARK II spherical analyzer with an aluminum anode (1486.6 eV) X-ray source, and the binding energy was calibrated by the C 1 $s$ peak (284.6 eV). Contact angle measurements were carried on an optical tensiometer (OCA 20, Dataphysics) using the sessile drop method with DI water. Solid state NMR measurements were performed on Bruker Avance III HD 400 MHz spectrometers using 3.2 mm magic-angle spinning probes. The XAFS spectra were recorded at room temperature using a 4-channel Silicon Drift Detector (SDD) Bruker 5040. Pd K-edge extended X-ray absorption fine structure (EXAFS) spectra were recorded in fluorescence mode. Temperature-programmed desorption (TPD) experiments were conducted with a TCD using 10 vol% $C_2H_2$ or $C_2H_4$ in Ar, respectively. In situ FTIR spectra of $H_2$ adsorbed on solid samples were collected on a Bruker Vector 70.

## Catalytic test

The process of selective hydrogenation was carried out in a 50 ml three-necked round bottom flask. Typically, 10 mg of catalyst and 1 mmol of substrate were dispersed in 10 ml of ethanol under constant stirring at 40 °C. A balloon filled with $H_2$ (1 bar) was then connected to the flask and purged for several times to remove any air before the reaction started. Samples were collected at regular intervals using syringe to monitor the reaction progress, and the product distribution was analyzed by a gas chromatography with a flame ionization detector (GC-FID), using octane as an internal standard. The product was further confirmed by gas chromatography-mass spectrometry (GC-MS). To ensure the reproducibility within ±0.5%, conversion and selectivity values were repeatedly measured under the same reaction conditions. For the hydrogen pretreatment experiment, hydrogen was introduced into the system for a certain period before adding the substrate, with all other conditions remaining unchanged.

The reaction rate was calculated using Eq. (1) and the conversion was controlled to be less than 20% to eliminate the effect of reverse reaction.

$$v = \frac{\Delta c}{\Delta t} = \frac{\Delta n}{V \Delta t} = \frac{n_0 - n_t}{V \Delta t} \tag{1}$$

## DFT calculations

Calculations were conducted using periodic, spin-polarized density functional theory (DFT) implemented in the Vienna ab initio program package (VASP), employing the projector augmented wave (PAW) method proposed by Blöchl[50] and implemented by Kresse[51]. A cutoff energy of 400 eV for plane waves was set throughout the calculations, and exchange-correlation functional approximation was treated with PBE functional. Gaussian electron smearing method with σ = 0.05 eV was used. A p (2 × 2) supercell with a 3-layer slab of 144 atoms for $WO_3$ (0 0 2) was modeled, and k-point of 2 × 2 × 1 was used for Brillouin zone sampling during structure optimization. During structural optimization, the bottom layer was fixed at a bulk truncated position, while the top three layers and the adsorbates were allowed to relax fully. The periodic condition was employed along the x and y directions and the vacuum region between the slabs was 15 Å, which was sufficiently large to keep spurious interactions negligible. The geometry optimization was stopped when the force residue on the atom was smaller than 0.02 eV and the energy difference was <10$^{-4}$ eV. The adsorption and dissociation energies for molecule chemisorption were defined as follows, respectively (Eq. 2)

$$E_{ads} = E_{tota} - E_{slab} - E_{mol} \tag{2}$$

where $E_{tota}$ is the total energy after a molecule adsorption on the catalyst, $E_{lsa}$ is the energy of the clean catalyst alone, and $E_{mol}$ is the energy of the molecule in the gas phase. Transition states (TS) were obtained using the nudged elastic band (NEB) method with a force convergence criterion of 0.05 eV/Å.

## Data availability

The data that support the findings of this study can be found in the manuscript and Supplementary information, or are available from the corresponding author upon request.

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

## Acknowledgements

Financial support from the National Key R&D Program of China (2021YFB3801600), the National Natural Science Foundation of China (22325204), and the "Pioneer" and "Leading Goose" R&D Program of Zhejiang (2023C01108 & 2022C01218 & 2022C01151) are greatly appreciated.

## Author contributions

J.X. performed the catalyst preparation, characterization, catalytic tests and calculations. Q.L., Y.L., and H. N. participated in the catalyst preparation and characterizations. Q.L., H.N., and B.L. provided helpful discussions. S.M. and Y.W. designed this study and analysed the data. J.X., S.M. and Y.W. wrote the manuscript. Y.W. supervised the project. All authors discussed the results and commented on the manuscript.

## Competing interests

The authors declare no competing interests.
