## [Peer Review File · Nature Communications]

Mediating trade-off between activity and selectivity in alkynes semi-hydrogenation via a hydrophilic polar layerREVIEWER COMMENTS

Reviewer #1 (Remarks to the Author):

This manuscript reports the synthesis of an array of ligand-free Pd-supported species on either WO₃ or MoWO₃ (Pd single atoms, SACs; and Pd NPs from 2.8 to 7.8 nm) and their use as catalysts in the semi-hydrogenation reaction of alkynes, particularly a representative propargyl alkyne. The results show that the catalyst is more selective (and practically as active) when the solid catalyst is pre-treated with H₂, and these results are explained here by the formation of a hydrophobic polar layer (HPL) on the catalyst surface due to the easy spillover of H atoms on the OH surface groups, further improved with Mo doping. In this way, a variety of alkynes can be selectively hydrogenated to alkenes, and also nitro to amine groups (Figure 6).

This Reviewer was sceptic when reading the first pages of the manuscript about the HPL and spillover explanations for the high selectivity found. However, after complete reading of the work, I think that these explanations are, at least, possible. However, other potential reasons for the findings found here still may apply, such as for instance that the formation of an amorphous shell on the crystalline solid after H₂ treatment or the formation of more vacancies on the support is behind the catalysis. This should be mentioned in the text. Nevertheless, the study here is very complete, consistent and original, mainly in the part regarding the HPL effect. Besides, this effect seems to occur for other hydrogenation reactions, supports and even other metals (see below). I think, thus, that the study is potentially publishable in Nat. Commun, after the following points are conveniently addressed:

- The title is completely different in the MS and SI. I have to say that I like more the title in the SI, the concept of "vertical" is difficult to visualize from an atomistic point of view.
- A critical point is the size of the Pd NPs and the absence of any Pd SAs (or other ultrasml Pd species such clusters) in the supported Pd NPs. It is difficult to assess these supported Pd species in the manuscript, since all microimages provided for Pd NPs-supported solids are at low magnification, having scale bars of 50-100 nm. In contrast, the Pd SAs are microimaged with much higher magnification. Why not the former? The fact that the synthetic procedure requires a calcination (to remove the stabilizing polymer) and re-hydrogenate back leaves doubt about the integrity of the original Pd NPs. The 5 and 8 nm Pd NPs should be visible in XRD, and the Scherrer equation should give an estimation of the size (double check with the TEM). Why Figure 2C does not show any Pd species?
- Planar Pd clusters will also fit the HPL effect as Pd SACs, since they do not have any vertical arrangement. I will leave a comment on this. Is the vertical effect linear, in other words, the "taller" NPs are less selective? Please explain. If not, are these results indirectly saying that the spillover effect only occurs as a monolayer?
- The catalytic activity is apparently very high, but measured as mol / (l · h). This parameter depends on the reaction concentration, thus it is not comparable between most systems. Indeed, a test was made here without solvent, which should give a much higher productivity value (not commented). In order to compare between catalysts, the TON or TOF values should be better employed. For instance, a TOF of >2 millions h⁻¹ has been recently reported for this reaction with PdCl₂ (J. Org. Chem. 2023, 88, 18–26). Please recalculate the activity values and compare with that reference and others.
- I miss experiments with D₂. For instance, the signals for the new O-D entities on the support will change in FT-IR and could be spotted easily. The KIE effect will also be determined.
- Figure 6, caption e, says "Pt". Was the reaction tested with Pt instead of Pd? There is not any comment in the text. This is relevant, and some characterization should be given for the Pt catalyst.
- The solid 1H NMR spectrum in Figure 2e shows a new signal at 1.1 ppm, which is assigned to new OH bonds. I do not agree with that, OHs tend to appear much higher, >3 ppm. Please comment.
- Figure numbers, captions and references in the text do not match in some cases (i.e. Figure 3c, Figures 2e-f). Please check.
- Experimental: A solution of PdCl₂ dissolved in water is used to prepare the Pd catalysts. PdCl₂ is hardly soluble in water (I am sure that not soluble at all at the specified concentrations), and it would

aggregate rapidly if dissolved. Please explain. H₂ is fed to the reaction with a balloon, is this system reliable when comparing reactions? The H₂ amount is always exactly the same?

- The concept of "breaking" LSRs is commented in the introduction but not further treated in the text. Please explain. How the present approach breaks the LSR?
- The most active supports are those having vacancies. Probably, more vacancies are generated with H₂ pre-treatment, how decoupling the vacancies and spillover effect? Are both directly related?
- Revise typos: Caption Fig 2 "Comparison", experimental "from Aladdin", "H₂". Fig. 6 caption "in a".

Reviewer #2 (Remarks to the Author):

It was a real pleasure to review this article by Yong Wang and his team. It has been shown that Palladium atoms isolated on an innovative support (Mo/WO₃) enable higher selectivity than those generally found in the literature for the selective hydrogenation of a wide range of alkynes to alkenes. This is a very interesting and innovative result, but it only appears at the end of the article and is not sufficiently developed.

It has also been shown that on a slightly different support (WO₃) the catalytic activity of isolated palladium atoms is higher than the one achieved by palladium nanoparticles (3-8 nm) for the same reaction. This is an original result that differs from the literature (it is generally accepted that this reaction is favoured with large nanoparticles (<https://doi.org/10.2533/chimia.2012.681>) or that SA/NP collaboration gives better results (<https://doi.org/10.1002/cctc.202300036>)) and is discussed in detail.

This work is highly original, but the experimental results are insufficient to validate the hypotheses. What's more, the choice of substrate, which eliminates the major challenge of alkyne hydrogenation, namely Cis/Trans selectivity, is detrimental to the impact of this article. Finally, the article is too long and needs to be simplified to make it easier to read. So as it stands I do not recommend its publication in nature communications.

To improve this article, I suggest the following:

- * Simplify and reorganise all the figures and adapt the titles. For example:

- Fig. 2 "Structural characterizations of Pd/WO₃" starts with a comparison of supports (2a ; 2b) not mentioned in the title.

- Fig. 3.d (f ?) and 4.a present results from the same series of experiments, each diluted with data from other manipulations.

- * Present the comparison of supports in terms of selectivity versus conversion (1st criterion for catalyst choice but not present in this version) and then activity. If possible, include Lindlar's reference catalyst in this comparison (which is mentioned as used but the results are not provided).

- * For the study of the activation of the Pd/WO₃ catalyst, add a continuous experiment that would enable the activation time to be compared under experimental conditions versus under gas phase pre-reduction. Alternatively, run several successive batches until the results stabilise. This would make it possible to decouple catalyst activation and the kinetic law. Indeed, for a low concentration of 0.1M it is highly unlikely to obtain a strongly negative apparent order as proposed in figure 3b. This hypothesis could also be evaluated by carrying out experiments at different concentrations.

- * Similarly this continuous study would enable a more reliable deactivation study to be carried out than that proposed in Fig. 5e which is unsatisfactory over only 10 cycles (TON < 35000 and conversion >95% see <https://doi.org/10.1021/acscatal.8b03199> for a better methodology).

- * The catalytic effect of the support discussed in Figs 3e (d?) and 3f is very interesting but the blank experiment with just the support is missing.

- * Simplify the scope of the paper by removing the nitro-aromatic hydrogenations.

- * For Fig. 3g explain how, with a homogeneous crystallite distribution, some nanoparticles can be charged with H₂ to dissociate H₂ and trigger the spillover while some others are charged with alkynes.

- * The TPDs (Fig 4d and 4e) show the effect of isolated atoms on adsorption equilibria. It would be interesting to compare with the Pd₁/MoWO₃ catalyst. A discussion citing the reference DOI:

- 10.1039/d1cy01016f which discusses in detail this topic would be a plus. Applying this methodology to

the new catalysts studied in this article would be of premium interest.

* I34: "The strong adsorption of alkyne can also result in self-poisoning of the catalyst" is new to me and not supported by any reference.

Reviewer #3 (Remarks to the Author):

Xiong et al. reported on a development of a catalyst for a selective hydrogenation of alkynes. A hydrophilic polar layer was induced via hydrogen spillover by choosing a reducible support for Pd atoms. The vertical effect of the polar layer guided the selectivity in the semihydrogenation of alkyne.

The development of a catalyst shows an elegant and novel design, that led to the targeted activity. The characterisation of the catalysts are thorough and comprehensive. The proposed mechanism is supported by the experimental data.

I believe that this manuscript deserves to be published in the Nature Communications journal.

Point-by-Point Response to the Reviewer's Comments

We would like to express our gratitude to the reviewers for their insightful and thorough feedbacks of our manuscript (NCOMMS-21-49916). Additionally, we extend our appreciation to the editor for handling our submission. We have diligently revised the manuscript in accordance with the reviewer's comments and suggestions, addressing each point individually. All the modifications have been highlighted in yellow within the revised manuscript. Below, we present our response to the referee's comments. We sincerely believe that the revised manuscript now meets the criteria for publication.

Detailed Response to Reviewer's Comments:

Referee #1:

General Comment: This Reviewer was sceptic when reading the first pages of the manuscript about the HPL and spillover explanations for the high selectivity found. However, after complete reading of the work, I think that these explanations are, at least, possible. However, other potential reasons for the findings found here still may apply, such as for instance that the formation of an amorphous shell on the crystalline solid after H₂ treatment or the formation of more vacancies on the support is behind the catalysis. This should be mentioned in the text. Nevertheless, the study here is very complete, consistent and original, mainly in the part regarding the HPL effect. Besides, this effect seems to occur for other hydrogenation reactions, supports and even other metals (see below). I think, thus, that the study is potentially publishable in Nat. Commun, after the following points are conveniently addressed.

Response: We would like to express our sincere thanks to the reviewer for the constructive and positive feedback. The patience and preciseness greatly helped us in enhancing this manuscript. To provide a clearer understanding of the rigorous experimental design, we have incorporated essential experimental details and supplemented the manuscript with additional experimental data to substantiate our

conclusions. In line with the reviewer's suggestions, we have meticulously addressed each point in a point-by-point response. Furthermore, we have taken care to refine the wording and descriptions pertaining to the spillover chemistry involved.

We are optimistic that these revisions will meet the reviewer's expectations, and we believe that the manuscript is now well-suited for publication.

Comment 1. The title is completely different in the MS and SI. I have to say that I like more the title in the SI, the concept of "vertical" is difficult to visualize from an atomistic point of view.

Response: We deeply apologize for the discrepancy in the titles between our manuscript and supplementary information (SI). In fact, we had been deliberating between these two titles for some time. After careful consideration, we are inclined to accept the reviewer's suggestion and will amend the title of our paper to "**Mediating trade-off between activity and selectivity in alkynes semi-hydrogenation via a hydrophilic polar layer.**" Thanks very much for the reviewer's valuable input.

Comment 2. A critical point is the size of the Pd NPs and the absence of any Pd SAs (or other ultrasmall Pd species such clusters) in the supported Pd NPs. It is difficult to assess these supported Pd species in the manuscript, since all microimages provided for Pd NPs-supported solids are at low magnification, having scale bars of 50-100 nm. In contrast, the Pd SAs are microimaged with much higher magnification. Why not the former? The fact that the synthetic procedure requires a calcination (to remove the stabilizing polymer) and re-hydrogenate back leaves doubt about the integrity of the original Pd NPs. The 5 and 8 nm Pd NPs should be visible in XRD, and the Scherrer equation should give an estimation of the size (double check with the TEM). Why Figure 2C does not show any Pd species?

Response: We appreciate the constructive comments from the reviewer. In response to the reviewer's suggestions, we conducted higher magnification microimaging of the Pd NPs. Regarding Pd/WO₃ (2.8 nm), the corresponding aberration-corrected HAADF-STEM image displayed in Fig. R1 clearly demonstrates the presence of Pd species in particle form, with no small clusters or single atoms observed. Additionally, high-

resolution TEM images of Pd/WO₃ with various particle sizes are provided in Fig. R2, revealing the absence of single atoms and clusters.

Regarding the XRD pattern, the absence of a metallic Pd peak may be attributed to the low Pd loading (references: Chem 2020, 6 (3), 752-765; ACS Catal. 2021, 11, 5666–5677; Appl. Catal. B: Environ. 2021, 299, 120648). To address the reviewer's concerns further, we prepared PdNP/WO₃ catalysts (7.8 nm) with higher loading (~2%). As shown in Fig. R3, the corresponding XRD shows a clear signal of metallic Pd, with the particle size calculated to be 8.9 nm according to the Scherrer equation, which is consistent with the TEM results. The absence of Pd nanoparticles in Fig. 2c can be attributed to the higher magnification and their relatively low distribution at the edges. Additionally, we have replaced the figure as shown in Fig. R4.

Fig. R1. Aberration-corrected HAADF-STEM image of Pd/WO₃ (2.8 nm). Scale bar: 5nm. (This figure has been included in the revised the supplementary information as Supplementary Fig. 10).

Fig. R2. TEM and HRTEM images of the Pd/WO₃ with different Pd particle sizes: (a,b,c) 2.8 nm, (d,e,f) 4.6 nm, (g,h,i) 7.8 nm. (Adopted from the Supplementary Fig.21 of the Supplementary information, respectively)

Fig. R3. XRD pattern of 1% Pd/WO₃ (7.8 nm) and WO₃.

Fig. R4. HRTEM of the fresh Pd/WO₃ and activated Pd/WO₃. Scale bar: 2 nm (Top) and 2 nm (bottom). (This figure has been included in the revised manuscript as Fig. 2c.)

Special changes are as follows:

Line 129-131 in the revised manuscript:

Aberration-corrected HAADF-STEM images indicated that the Pd species existed in the form of particles, with no small clusters or single atoms observed (Supplementary Fig. 10).

Comment 3. Planar Pd clusters will also fit the HPL effect as Pd SACs, since they do not have any vertical arrangement. I will leave a comment on this. Is the vertical effect linear; in other words, the “taller” NPs are less selective? Please explain. If not, are these results indirectly saying that the spillover effect only occurs as a monolayer?

Response: We greatly appreciate the insightful comments provided by the reviewer. After a thorough analysis of the relevant experimental data, we have made several noteworthy observations.

Firstly, as depicted in Figure 3d in the revised manuscript, we have observed that as the size of Pd particles increases, their intrinsic reaction activity gradually decreases, demonstrating a certain degree of linear correlation. We believe that this phenomenon is related to the dispersion of Pd metal; as particle size increases, dispersion decreases, leading to reduced intrinsic activity.

However, when examining selectivity (Figure 4a), there is no apparent linear relationship between selectivity and particle size. Only single-atom catalysts exhibit high selectivity. In the case of single-atom catalysts, we have observed that both activity and selectivity increase gradually with prolonged hydrogen treatment time (positively correlated with the surface hydroxyl content), but after a certain period, there is no significant change.

From our experimental data, it is evident that for nanoparticles (NPs), selectivity is not significantly influenced by hydrogen spillover (Figure 4a). Therefore, there is no issue of "taller" NPs being less selective. Based on these results, we can infer that the spillover effect only occurs as a monolayer.

Regarding planar Pd clusters, theoretically, they are also influenced by hydrogen spillover. We hypothesize that this influence is related to the size of the planar Pd clusters themselves, as illustrated in Scheme R1. When the Pd planar surface is relatively large, the impact of hydrogen spillover is diminished, and selectivity is compromised.

We hope that these observations provide valuable insights into the relationship between particle size, hydrogen spillover, and selectivity in our study.

Scheme R1. Schematic illustration of the hydrogenation behavior of planar Pd cluster.

Comment 4. The catalytic activity is apparently very high, but measured as $\text{mol}/(\text{l} \cdot \text{h})$. This parameter depends on the reaction concentration, thus it is not comparable between most systems. Indeed, a test was made here without solvent, which should give a much higher productivity value (not commented). In order to compare between catalysts, the TON or TOF values should be better employed. For instance, a TOF of >2 millions h^{-1} has been recently reported for this reaction with PdCl_2 (*J. Org. Chem.* 2023, 88, 18–26). Please recalculate the activity values and compare with that reference and

others.

Response: We appreciate the valuable comments from the reviewer regarding catalytic activity. To facilitate a comprehensive comparison of catalytic activities among different catalysts, we have also calculated and compared the catalytic performance normalized to the number of Pd sites, which can be regarded as a form of TOF. This comparative analysis is presented in Figure R6 (adapted from Fig. 5e of the manuscript). Specifically, the specific activity normalized by Pd for activated Pd1/MoWO₃ is approximately 10,200 h⁻¹ under conditions of 40 °C and 0.1 MPa hydrogen. In contrast, the literature reference reports a TOF of >2 million h⁻¹ using a homogeneous Pd catalyst (β-PdCl₂) under more stringent reaction conditions of 90°C and 0.5 MPa hydrogen. However, it is noteworthy that when the temperature is controlled at 30 °C, the TOF value significantly decreases to 4,500 h⁻¹ (J. Org. Chem. 2023, 88, 18–26).

Furthermore, it's important to emphasize that the hydrogenation of alkynes in the literature reference is conducted using a homogeneous Pd catalyst (β-PdCl₂). Such homogeneous systems pose challenges for catalyst recovery and are susceptible to Pd species leaching during the reaction, resulting in deactivation.

Lastly, it should be noted that the PdCl₂ catalyst mentioned in the literature cannot be utilized under solvent-free conditions. This limitation arises from the requirement for a solvent to facilitate the formation of Pd⁰ active species, and a sufficient quantity of solvent is necessary to ensure the proper dispersion of the Pd species. These limitations restrict the applicability of such catalysts in industrial manufacturing. In contrast, our Pd1/MoWO₃ catalyst demonstrates excellent reactivity under solvent-free conditions, as depicted in Figure R6, with a TOF of 88,000 h⁻¹, which is 11 times higher than that of the commercial Lindlar catalyst (8,000 h⁻¹).

We hope these explanations provide a clear understanding of the considerations we have taken into account when evaluating the catalytic activity in our study.

Fig. R5. Comparison of specific activity normalized by Pd and MBE selectivity for fresh Pd₁/WO₃, activated Pd₁/WO₃, fresh Pd₁/MoWO₃, activated Pd₁/MoWO₃ and other catalysts toward semi-hydrogenation of MBY. (Adopted from Fig.5d of the manuscript)

Fig. R6. Catalytic activity and selectivity for the selective hydrogenation of MBY over activated Pd₁/MoWO₃ and Lindlar catalyst without solvents. Reaction condition: 1.5×10^{-2} wt % and 4.2×10^{-2} wt % of Pd for Pd₁/MoWO₃ and Lindlar catalyst, 5 mL MBY, 80 °C, 1 bar H₂, 1000 rpm.

Comment 5. I miss experiments with D₂. For instance, the signals for the new O-D entities on the support will change in FT-IR and could be spotted easily. The KIE effect will also be determined.

Response: Thanks for the valuable comment. Following the reviewer's suggestion, we conducted experiments using D₂. Initially, we employed in situ Fourier transform infrared spectroscopy (FT-IR) to detect the activated D₂ species on Pd/WO₃. As depicted in Figure R7, upon the introduction of D₂, several absorption peaks emerged in the range of 2700~2850 cm⁻¹, corresponding to the stretching vibration of O-D entities on the support. This observation demonstrates the occurrence of hydrogen spillover.

Furthermore, we also investigated the kinetic isotope effect (KIE) on activated Pd/WO₃ (Pd/WO₃-D) by using D₂ instead of H₂, as shown in Figure R8. In the semi-hydrogenation of MBY process, the calculated KIE value ($k_H/k_D = 1.64$) exhibits a typical secondary isotope effect. This result indicates that the rate-limiting step is the MBY hydrogenation rather than H₂ activation, providing further evidence that hydrogen spillover on the hydrophilic polar layer (HPL) effectively mitigates the poisoning effect of MBY.

Fig. R7. In situ FT-IR spectra of Pd/WO₃ (2.8 nm) in a flow of H₂ and D₂. (This figure has been included in the revised supplementary information as Supplementary Fig. 14).

Fig. R8. Isotope effect in MBY semi-hydrogenation catalyzed by activated Pd/WO₃ (2.8 nm), showing a typical secondary isotope effect. (This figure have been included in the revised supplementary information as Supplementary Fig. 18)

Special changes are as follows:

Line 138-140 in the manuscript:

In situ H₂ infrared spectroscopy showed that the peaks for stretching vibration of W-OH at 3290 cm⁻¹ emerged and enhanced with increasing hydrogen exposure time (Fig. 2d and Supplementary Fig. 13), while switching the atmosphere from H₂ to D₂, the signals for the new O-D entities on the support could be spotted easily (Supplementary Fig. 14), which demonstrated the occurrence of hydrogen spillover^{43,44}.

Line 168-172 in the manuscript:

In addition, the kinetic isotope effect (KIE) was explored on activated Pd/WO₃ (Pd/WO₃-H) using D₂ instead of H₂ (Supplementary Fig. 18). The calculated KIE value ($k_H/k_D = 1.64$) indicates that the rate-limiting step is MBY hydrogenation rather than H₂ activation, further proving that the HPL effectively reduced the poisoning effect of MBY.

Comment 6. Figure 6, caption e, says "Pt". Was the reaction tested with Pt instead of Pd? There is not any comment in the text. This is relevant, and some characterization

should be given for the Pt catalyst.

Response: Thanks for the reviewer's valuable comment. The hydrogenation of substituted nitrobenzenes has indeed been tested with Pt instead of Pd. We have provided some characterizations corresponding to the Pt catalyst. Please note that the Pt signal of EXAFS and XPS may be severely distorted due to the close atomic numbers of Pt (78) and W (74), making it difficult to obtain reliable information. Therefore, we have included only the corresponding aberration-corrected HAADF-STEM (Fig. R9), EDS mapping (Fig. R10), and XRD pattern (Fig. R11).

Fig. R9. Aberration-corrected HAADF-STEM image of Pt₁/MoWO₃, with single-site Pd marked by red circles. (This figure has been included in the revised the supplementary information as Supplementary Fig. 52)

Fig. R10. EDX mapping images of different elements of Pt₁/MoWO₃. (This figure has been included in the revised the supplementary information as Supplementary Fig. 53)

Fig. R11. XRD pattern of Pt₁/MoWO₃. (This figure has been included in the revised the supplementary information as Supplementary Fig. 54)

Comment 7. The solid ¹H NMR spectrum in Figure 2e shows a new signal at 1.1 ppm, which is assigned to new OH bonds. I do not agree with that, OHs tend to appear much higher, >3 ppm. Please comment.

Response: Regarding the solid ¹H NMR spectrum in Figure 2e, it shows two forms of -OH groups on the catalyst surface. One is the bridging hydroxyl (as you mentioned, typically appearing at positions >3 ppm), which is more acidic. The other is the terminal hydroxyl (less acidic), usually located around 1 ppm. These conclusions can be referenced in the following literature: (Int. J Refract. Met. H. 1998, 16, 23-30; J. Phys. Chem. B 2006, 110, 10662-10671; J. Am. Chem. Soc. 2021, 143, 9236-9243; Nat Commun. 2023, 14, 4209).

Comment 8. Figure numbers, captions and references in the text do not match in some cases (i.e. Figure 3c, Figures 2e-f). Please check.

Response: Thanks for the reviewer's comment. We have made the necessary revisions to the figure numbers, captions, and references in the text as suggested.

Comment 9. Experimental: A solution of PdCl₂ dissolved in water is used to prepare the Pd catalysts. PdCl₂ is hardly soluble in water (I am sure that not soluble at all at the specified concentrations), and it would aggregate rapidly if dissolved. Please

explain. H₂ is fed to the reaction with a balloon, is this system reliable when comparing reactions? The H₂ amount is always exactly the same?

Response: Thanks for the reviewer's valuable comment. Firstly, we apologize for the confusing expression regarding the PdCl₂ solution. In actuality, the PdCl₂ solution is prepared using a dilute hydrochloric acid solution to ensure proper dissolution. We have revised the sentences in the experimental section accordingly.

Additionally, the introduction of hydrogen gas through the balloon serves to balance the gas pressure and maintain a constant H₂ pressure of 1 bar in the closed system. It's important to note that the consumption of hydrogen is minimal since the substrate used in the reaction is only 1 mmol. Therefore, the overall pressure of the system is not significantly affected. We conducted multiple repeated experiments to ensure the reliability of the system when comparing reactions, as shown in Figure R12. The results indicate that the average error of each reaction falls within an acceptable range.

It's worth mentioning that this method of hydrogenation conducted at ambient pressure is employed by many works, as referenced (J. Mater. Chem. A, 2019, 7, 4714-4720; Mater. Horiz., 2017, 4, 584-590; Molecular Catalysis 2020, 488, 110923; CCS Chemistry 2019, 1 (2), 207-214; Catal. Sci. Technol., 2021, 11, 4539-4548; Angew. Chem. 2022, 134, e202202923; J. Catal. 2017, 350, 13-20; Nat Commun 2022, 13, 2754).

Fig. R12. (a) Parallel experiment of selective hydrogenation of MBY over activated Pd/WO₃. (b) Corresponding error bars of the kinetic hydrogenation curves.

Special changes are as follows:

Line 373-377 in the manuscript:

Next, 100 mg of PdCl₂ was initially dissolved in 10 ml of hydrochloric acid solution (0.6 M) to obtain a PdCl₂ solution. Subsequently, 150 μL of the PdCl₂ hydrochloric acid solution (containing 6 mg·ml⁻¹ Pd) was diluted with 8 ml of deionized water and added to the WO₃ dispersion. The mixture was then stirred continuously for 1 h.

Comment 10. The concept of “breaking” LSRs is commented in the introduction but not further treated in the text. Please explain. How the present approach breaks the LSR?

Response: We appreciate the valuable comments provided by the reviewer and apologize for the lack of clarity in our description regarding the disruption of linear scaling relationships (LSRs). We are pleased to offer a detailed explanation.

LSRs refer to linear correlations between the bond energies of various surface-adsorbed species. In simpler terms, changes in the electronic structure of active sites simultaneously influence the adsorption of both alkenes and alkynes (as discussed in Nat. Commun. 2022, 13(1): 2754 and Science, 2008, 320(5881): 1320-1322). Consequently, the pursuit of improved selectivity often comes at the expense of catalytic activity, as demonstrated in ACS Appl. Mater. Interfaces 2021, 13, 31775-31784, and Chem 2018, 4, 1080-1091. The challenge lies in designing a highly efficient catalyst that exhibits both exceptional activity and selectivity.

In our current work, we introduce a novel strategy for constructing a hydrophilic polar layer (HPL) through hydrogen spillover. This approach offers the advantage of minimally altering the intrinsic activity of the metal active sites while enhancing selectivity. The HPL achieves this without directly modifying the electronic structure of the active sites but by instead modifying the surrounding surface microenvironment. The presence of the HPL introduces a polarity repulsion effect between the HPL and unsaturated carbon-carbon bonds, thereby weakening the adsorption of alkenes and alkynes on active sites. It is important to note that alkenes experience stronger repulsion compared to alkynes due to their larger steric hindrance and weaker polarity. This

selective repulsion prevents the over-hydrogenation of as-formed alkenes. Consequently, this unique approach to selective regulation effectively enhances alkene selectivity without altering the properties of the active sites (in this case, Pd single atoms), thus disrupting the linear scaling relationships. Furthermore, the presence of a hydrogen reservoir, which involves activated hydrogen on the support, further enhances catalytic activity.

In summary, our approach of constructing HPL effectively addresses the trade-off between catalytic activity and selectivity through the utilization of hydrogen spillover. We have incorporated this discussion into the revised manuscript for further clarity.

Special changes are as follows:

Line 162-167 in the manuscript:

With an extension of the pretreatment time, the reaction order of MBY gradually decreased to -0.25. This observation indicates that the hydrophilic polar layer (HPL) formed through hydrogen spillover can effectively alleviate the self-inhibition of MBY without directly altering the electronic structure of the active Pd sites (see Supplementary Fig. 17), consequently leading to an enhancement in activity.

Line 272-277 in the manuscript:

As a result, the unique selectivity promotion phenomenon with hydrogen spillover is observed only in the case of WO₃-supported single-atom Pd catalysts in alkyne semi-hydrogenation. The approach of constructing an HPL around the Pd single atom effectively regulates the adsorption of alkenes and alkynes without altering the electronic structure of the active sites. This successful disruption of the linear scaling relationships (LSRs) leads to an enhancement in selectivity without compromising the catalyst's intrinsic activity. It should be noted that the effect of the HPL is also applicable to other single atom catalysts with reducible supports where hydrogen spillover occurs.

Comment 11. The most active supports are those having vacancies. Probably, more vacancies are generated with H₂ pre-treatment, how decoupling the vacancies and

spillover effect? Are both directly related?

Response: We appreciate the valuable comment from the reviewer. If the improvements in both activity and selectivity are ascribed to the influence of vacancies, there should, in theory, be no distinction between single-atom catalysts and nanoparticle catalysts. In actuality, the discrepancies in selectivity between single-atom catalysts and nanoparticle catalysts can be elucidated by the preeminent role played by the spillover effect.

Prior studies have demonstrated that, in the absence of potent reducing agents, the creation of oxygen vacancies necessitates more stringent conditions, such as elevated temperature and pressure within the reducing atmosphere (as discussed in Chem. Eng. J. 2023, 454, 14037; Catal. Rev. 2022, 1-54; Adv. Funct. Mater. 2022, 32, 2109503). Consequently, the generation of oxygen vacancies in Pd/WO₃ during the hydrogen pretreatment process should be inconsequential due to the mild conditions employed (i.e., 40°C at atmospheric pressure). This is further corroborated by the Electron Paramagnetic Resonance (EPR) spectra (see Fig. R13), where the signal corresponding to oxygen vacancies displays negligible changes after activation.

It is noteworthy that a substantial formation of oxygen vacancies typically leads to alterations in the crystal phase of WO₃, resulting in species such as WO_{2.9} or WO_{2.72} (as observed in J. Mater. Chem. A, 2018, 6, 6780-6784; J. Catal. 2021, 402, 208-217; Appl. Catal. B: Environ. 2023, 324, 122202). However, the X-ray Diffraction (XRD) pattern following H₂ activation, as shown in Fig. R14, demonstrates that the bulk phase structure of WO₃ remains unchanged.

Furthermore, it is evident that the Pd/WO₃ catalyst before activation already possesses a certain quantity of oxygen vacancies resulting from the hydrothermal synthesis process. This presence of oxygen vacancies has already been manifested in the fresh catalyst and has proved unsatisfactory in terms of both activity and selectivity. Based on these observations, the enhanced performance of the activated Pd/WO₃ catalyst cannot be attributed to the contribution of oxygen vacancies. Considering that the quantity of oxygen vacancies does not significantly change after the hydrogen

pretreatment process, while various characterizations indicate an increase in the number of hydroxyl groups on the catalyst surface, we can conclude that hydroxyl groups, or in other words, the hydrogen spillover process, have a more pronounced impact on the reaction performance.

Fig. R13. EPR spectra for fresh and activated Pd/WO₃.

Fig. R14. XRD patterns for fresh Pd/WO₃ and activated Pd/WO₃. (Adopted from Supplementary Fig. 12 of the Supplementary information)

Comment 12. Revise typos: Caption Fig 2 “Comparison”, experimental “from Aladdin”, “H2”. Fig. 6 caption “in a”.

Response: Thanks. We have corrected the typos error accordingly.

Referee #2:

***General Comment:** It was a real pleasure to review this article by Yong Wang and his team. It has been shown that Palladium atoms isolated on an innovative support (Mo/WO₃) enable higher selectivity than those generally found in the literature for the selective hydrogenation of a wide range of alkynes to alkenes. This is a very interesting and innovative result, but it only appears at the end of the article and is not sufficiently developed.*

It has also been shown that on a slightly different support (WO₃) the catalytic activity of isolated palladium atoms is higher than the one achieved by palladium nanoparticles (3-8 nm) for the same reaction. This is an original result that differs from the literature (it is generally accepted that this reaction is favoured with large nanoparticles (<https://doi:10.2533/chimia.2012.681>) or that SA/NP collaboration gives better results (<https://doi.org/10.1002/cctc.202300036>)) and is discussed in detail.

This work is highly original, but the experimental results are insufficient to validate the hypotheses. What's more, the choice of substrate, which eliminates the major challenge of alkyne hydrogenation, namely Cis/Trans selectivity, is detrimental to the impact of this article. Finally, the article is too long and needs to be simplified to make it easier to read. So as it stands I do not recommend its publication in nature communications.

Response: We extend our heartfelt gratitude to the reviewer for her/his patience and precision, which have greatly contributed to enhancing the quality of our manuscript. With regard to the concerns the reviewer raised, we have provided a detailed point-by-point response as outlined below.

Concerns:

Q1: The Pd₁/MoWO₃ exhibits great catalytic performance for the selective hydrogenation of a wide range of alkynes to alkenes, but only a brief description of the catalyst is provided at the end of the article without further sufficiently exploration.

Response: The primary objective of this article is to introduce a strategy for creating a hydrophilic polar layer (HPL) through hydrogen spillover, aimed at enhancing the

selectivity of alkenes while preserving the intrinsic activity of the metal active sites. This strategy effectively overcomes the limitations posed by linear scaling relationships (LSRs) and resolves the challenge of balancing activity and selectivity in alkyne semi-hydrogenation. Another significant aspect of this research is the discovery of the size effect of hydrogen spillover on catalyst selectivity regulation.

The mechanistic elucidation in this study is relatively intricate, necessitating the simplification of the catalyst structure as much as possible for investigation. While Pd1/MoWO₃ exhibits superior selectivity compared to what is typically reported in the literature for the selective hydrogenation of a wide range of alkynes to alkenes, its more complex catalyst structure, in contrast to Pd/WO₃, introduces additional factors that complicate the interpretation of the reaction mechanism. Therefore, our main focus in this work is on Pd/WO₃ as the primary research subject.

Nonetheless, experiments related to Pd/MoWO₃ have been conducted to demonstrate the universality of the HPL strategy and to optimize catalytic performance. Further exploration of Pd/MoWO₃ can be undertaken as a distinct research endeavor.

Q2: The major challenge of alkyne semi-hydrogenation, namely Cis/Trans selectivity, is not investigated under the choice of terminal alkyne substrate

Response: In the process of alkyne semi-hydrogenation, several side reactions occur, including alkene over-hydrogenation, isomerization (particularly cis-trans isomerization), and the coupling of unsaturated hydrocarbons. Achieving high selectivity for the desired alkene product is challenging due to these side reactions. Alkynes can typically be classified into terminal alkynes, internal alkynes, and acetylenes. In the semi-hydrogenation of acetylenes, major issues include the coupling of unsaturated molecules and the over-hydrogenation of ethylene. Internal alkynes face problems like cis-trans isomerization and the over-hydrogenation of alkenes. In the case of terminal alkynes (the substrates chosen in this work), the issue of cis-trans isomerization is eliminated, but they are prone to more severe over-hydrogenation reactions compared to internal alkynes due to their lower steric hindrance. Over-hydrogenation of alkenes is a common problem for all alkyne substrates and represents

Fig. R1. Selective hydrogenation of alkynes and other substrates over Pd₁/MoWO₃-H. The red color represented that the C≡C bonds were hydrogenated to corresponding C=C. ^aReaction conditions: 10 mL of ethanol, 40 °C, 1 bar H₂, and 1000 rpm. 1 mmol substrate, 0.028 mol% Pd. ^b60 °C. ^c80 °C, 0.084 mol% Pd. ^d0.056 mol% Pd, 1 Mpa H₂. ^e 0.056 mol% Pt. (This figure has been updated in the revised manuscript as Fig. 6.)

the most significant side reaction in this process. Most research in this area focuses on addressing this issue, and achieving high selectivity for the hydrogenation of terminal alkynes remains a significant challenge. As a result, we conducted the semi-hydrogenation of 2-methyl-3-butyn-2-ol, one of the typical terminal alkynes, which is a crucial intermediate for producing vitamins and spices, as a probe reaction.

Furthermore, while trans-alkenes are thermodynamically more stable than cis-alkene isomers, heterogeneous metal catalysts, such as Lindlar's catalysts, tend to favor the formation of cis-alkenes due to the preferred Horiuti–Polanyi mechanism. In our work, the semi-hydrogenation of internal alkyne substrates included in the expansion of substrate scope (Fig. R1) all exhibits excellent selectivity towards cis-alkenes. However, achieving trans-selective hydrogenation of internal alkynes is highly challenging, and suppressing the over-hydrogenation of the resulting trans-alkene is a prerequisite for addressing this issue. We are indeed concerned about the problem of trans-selectivity in the hydrogenation of internal alkynes. Our RhTi/SiO₂ catalyst currently achieves 73% trans-selectivity, and we remain committed to making further progress in this area.

Q3: The article is too long and needs to be simplified to make it easier to read

Response: We have implemented the required revisions in response to your valuable feedback and the constructive comments provided by the reviewer. These revisions have significantly improved the content of our manuscript, and we believe that it is now well-suited for publication. The specific changes will be outlined below.

Detailed Comments:

Comment 1. Simplify and reorganise all the figures and adapt the titles. For example: Fig. 2 “Structural characterizations of Pd/WO₃” starts with a comparison of supports (2a ; 2b) not mentioned in the title.

Fig. 3.d (f ?) and 4.a present results from the same series of experiments, each diluted with data from other manipulations.

Response: Thank you for your valuable comment, which we have taken into

consideration. As a result, we have made adjustments to the title to ensure accuracy in our expression. We fully appreciate the reviewer's concern regarding the organization of the figures, and we are pleased to provide an explanation.

Indeed, the results presented in Fig. 3d and Fig. 4a originate from the same series of experiments. However, we chose to present them as separate figures, each corresponding to selectivity and activity, for specific reasons. This series of experiments involved a total of eight individual tests, covering catalytic hydrogenation performance over fresh and activated Pd/WO₃ with different sizes of Pd NPs (2.8 nm, 4.6 nm, and 7.8 nm) and single atoms. Our intention in dividing them into two figures was to avoid overcrowding and to make it easier to provide a clear performance comparison. Trying to incorporate all the selectivity and activity data from these eight experiments into a single figure would have made it too cluttered and challenging to present a meaningful performance contrast.

Additionally, each figure in the manuscript is organized around a specific topic. For example, Figure 2 focuses on the comparison of catalysts with different supports before and after hydrogen activation, as well as the evidence and effects of hydrogen spillover on the Pd/WO₃ catalyst. Figure 3 emphasizes the activity comparison of Pd/WO₃ at different scales and delves into the underlying mechanism studies, such as the contribution of the support and the discovery of hydrogen pools. Figure 4 centers on the selectivity comparison of Pd/WO₃, where we unveil unconventional selectivity enhancement in the single-atom catalysts and explore the size effect of the hydrophilic polar layer (HPL). The goal is to ensure that each figure addresses a specific aspect of our research.

In our opinion, both Fig. 3d and Fig. 4a are pivotal data within their respective figure sets. Reorganizing them into a single figure could disrupt the coherence and create additional challenges for readers. We have considered the feedback carefully and believe that the current arrangement provides the best clarity and understanding of our findings.

Special changes are as follows:

Line 674-675 and 733-734 in the manuscript:

Fig. 2 | Comparison of catalytic performance and structural characterizations of Pd/WO₃

Line 683 and 744 in the manuscript:

Fig. 3 | Mechanistic analysis of activity enhancement in Pd/WO₃ Catalyst

Line 697-698 and 759-760 in the manuscript:

Fig. 4 | Mechanistic analysis of selectivity enhancement in Pd₁/WO₃ Catalyst

Line 708-709 and 771-772 in the manuscript:

Fig. 5 | Influence of Mo doping on the structural properties and catalytic performance of Pd₁/MoWO₃.

Line 721-722 and 785-786 in the manuscript:

Fig. 6 | The application of the Pd₁/MoWO₃-H catalyst in various alkynes and other substrates

Comment 2. Present the comparison of supports in terms of selectivity versus conversion (1st criterion for catalyst choice but not present in this version) and then activity. If possible, include Lindlar's reference catalyst in this comparison (which is mentioned as used but the results are not provided).

Response: Thanks for the valuable input from the reviewer. Following the reviewer's suggestion, we have included a comparison of supports based on selectivity versus conversion and then activity, as illustrated in Figures R2 to R4. It's worth noting that activated Pd/WO₃ exhibits exceptional activity and selectivity, with the most significant enhancement in activity observed after hydrogen pretreatment. As a result, we have chosen WO₃ as the primary focus of our research in this study. While the selectivity of TiO₂ is on par with WO₃, the improvement in its activity is not as pronounced. Additionally, the selectivity of activated Pd₁/TiO₂ is not as high as that of Pd₁/WO₃, so we have not pursued it further. Furthermore, we have included a comparison of the catalytic performance of the Lindlar catalyst for reference in Figure R5.

Fig. R2. Comparison for the catalytic activity of the MBY hydrogenation with and without hydrogen pretreatment over reducible supports (a) and irreducible supports (b). (Adopted from Fig. 2a and 2b of the revised manuscript).

Fig. R3. Comparison for the selectivity versus conversion of the MBY hydrogenation with and without hydrogen pretreatment over reducible supports (a) and irreducible supports (b). (This figure has been included in the revised the supplementary information as Supplementary Fig. 7)

Fig. R4. Comparison for the activity of the MBY hydrogenation with and without hydrogen pretreatment over reducible supports (a) and irreducible supports (b). (This figure has been included in the revised the supplementary information as Supplementary Fig. 6).

Fig. R5. Catalytic activity and selectivity for the selective hydrogenation of MBY over commercial Lindlar catalyst. (This figure has been included in the revised the supplementary information as Supplementary Fig. 48).

Special changes are as follows:

Line 122-128 in the manuscript:

No detectable variation in selectivity was observed among these Pd NPs catalysts, whether they are activated or not (Supplementary Fig. 7). This difference in activity strongly suggests that hydrogen spillover plays a significant role in the hydrogenation

process⁴¹. Considering that Pd/WO₃ showed the most significant increase in catalytic activity (4-fold) and relatively high selectivity, we chose WO₃ as the support to explore the underlying mechanism behind this observed activity enhancement.

Lin 311 in the manuscript:

the activated Pd₁/MoWO₃-H catalyst demonstrated an impressive yield of MBE up to 98.4% and a 26-fold activity (10200 h⁻¹ at 298 K) increase compared to commercial Lindlar catalysts for the semi-hydrogenation of MBY (Supplementary Fig. 48).

Comment 3. For the study of the activation of the Pd/WO₃ catalyst, add a continuous experiment that would enable the activation time to be compared under experimental conditions versus under gas phase pre-reduction. Alternatively, run several successive batches until the results stabilise. This would make it possible to decouple catalyst activation and the kinetic law. Indeed, for a low concentration of 0.1M it is highly unlikely to obtain a negative apparent order as proposed in figure 3b. This hypothesis could also be evaluated by carrying out experiments at different concentrations.

Response: Thanks for the valuable comment from the reviewer. Following the reviewer's suggestion, we conducted several successive batch experiments. As shown in Fig. R6, the catalytic activity significantly improved in the second reaction and remained relatively stable thereafter, which indicates that Pd/WO₃ can also be in-situ activated during the hydrogenation without hydrogen pretreatment. Additionally, by extending the pretreatment time, one can observe that the catalytic activity gradually increased with the activation time and reached a stable state after 30 minutes, indicating that the catalyst is fully activated by then (Fig. R7a).

Regarding the relationship between catalyst activation and kinetics, the concentration of substrates at the inflection point of the kinetic behavior can be regarded as the critical inhibitory concentration (CIC), beyond which the hydrogenation process would be restrained by competitive adsorption. With the extension of the pretreatment time, the CIC gradually reduced, along with the decreased reaction order of MBY. Furthermore, the curve of activation time versus intrinsic activity clearly demonstrates that the catalytic activity increases with the activation degree of Pd/WO₃ (Fig. R7b).

Regarding the negative apparent order of MBY in this reaction, we conducted the hydrogenation at different concentrations as per the reviewer's suggestions. Fig. R8 shows that the reaction order of MBY remains negative with a value of -0.76, which indicates that the adsorption of MBY on the active sites is excessively strong. This result is supported by the computational results in the present work, showing that the adsorption energy of MBY is -1.21 eV on Pd single atoms. Previous studies have also demonstrated that the adsorption energy of MBY on Pd (111) is higher than 1.50 eV (Catal. Sci. Technol., 2021, 11, 6205-6216; ACS Appl. Mater. Interfaces 2021, 13, 31775–31784; Angew. Chem. Int. Ed. 2022, e202202923). In most cases, alkyne molecules can bind much more strongly to the catalyst's surface due to the specific property of unsaturated triple bonds, resulting in high coverage of adsorbed alkyne on active sites (Appl. Catal. 1983, 6, 41–51; Appl. Catal. 1985, 15, 317–326; Nano Res. 2022, 15, 10044–10062). The excessive adsorption of alkyne will block active sites and inevitably weaken hydrogen dissociation due to competitive absorption, which leads to such a negative reaction order of MBY.

Fig. R6. Successive batch experiments of MBY semi-hydrogenation on Pd/WO₃ (2.8 nm). Reaction condition: 10 ml of ethanol, 40 °C, 1 bar H₂, and 1000 rpm. 1 mmol substrate, 10 mg catalyst. (This figure has been included in the revised the supplementary information as Supplementary Fig. 16).

Fig. R7. (a) The catalytic performance and (b) corresponding reaction rate of MBY semi-hydrogenation on Pd/WO₃ (2.8 nm) after pre-activation in H₂ for different period. Reaction condition: 10 ml of ethanol, 40 °C, 1 bar H₂, and 1000 rpm. 1 mmol substrate, 10 mg catalyst. (This figure has been included in the revised the supplementary information as Supplementary Fig. 15).

Fig. R8. Reaction orders of MBY calculated on Pd/WO₃ (2.8 nm) evaluated by carrying out experiments at different concentrations.

Special changes are as follows:

Line 154-158 in the manuscript:

Successive batch experiments demonstrated that the activity significantly improved in

the second run and remained stable thereafter, indicating that Pd/WO₃ could also be in-situ activated during hydrogenation without the need for hydrogen pretreatment (Supplementary Fig. 16).

Comment 4. Similarly this continuous study would enable a more reliable deactivation study to be carried out than that proposed in Fig. 5e which is unsatisfactory over only 10 cycles (TON < 35000 and conversion >95% see <https://doi.org/10.1021/acscatal.8b03199> for a better methodology).

Response: Thank you for this valuable suggestion. We have thoroughly reviewed the deactivation study methodology described in the referenced work (ACS Catal. 2018, 8, 9, 8597–8599), and the relevant method is detailed in Fig. R6.

Fig. R9. Assessment of catalyst stability under batch conditions: (a) invalid assessment by recycle: the amount of catalyst may be far more than needed to achieve full conversion in the allotted time, masking the presence of deactivation; (b) valid assessment by rates: the change in the apparent rate constant in the consecutive experimental runs 1–3 serves to quantify the extent of deactivation. (Adopted from Figure 1 in the ACS Catal. **2018**, 8, 9, 8597–8599)

For the cycling stability test of Pd₁/MoWO₃ presented in Fig. 5e, **it's important to note that the termination of each cycle is controlled at 85%-90% conversion rather than 100%, especially in the first cycle.** This approach is employed to avoid issues mentioned in Fig. R9a, where the amount of catalyst may far exceed what is

required for full conversion within the specified time, potentially masking the presence of deactivation. Furthermore, in Fig. R10, we provide a comparison of each cycle in terms of conversion versus time curves, which serves as a valid assessment, as shown in Fig. R9b. It is evident from these figures that the catalyst undergoes an activation process in the first cycle, and from the second cycle onwards, the catalytic activity remains stable. The apparent reaction rate hardly changes in the consecutive experimental runs 2–11, indicating excellent cyclic stability of the catalyst.

Fig. R10. Comparison of each cycle in terms of conversion versus time curves on Pd₁/MoWO₃. Reaction condition: 10 ml of ethanol, 40 °C, 1 bar H₂, and 1000 rpm. 4 mmol substrate, 40 mg catalyst. (This figure has been included in the revised the supplementary information as Supplementary Fig. 49).

Comment 5. The catalytic effect of the support discussed in Figs 3e (d?) and 3f is very interesting but the blank experiment with just the support is missing.

Response: Thanks. It's important to clarify that Figures 3d and 3f primarily investigate the effect of different Pd loadings on activity with varying particle sizes. In each figure, the horizontal axis represents the Pd loading, which makes it challenging to include the activity of the pure support for direct comparison. Furthermore, it's worth noting that the pure WO₃ support lacks activity for the selective hydrogenation of alkynes due to its limited ability for hydrogen dissociation, and hydrogen spillover effects are unlikely to occur on the pure WO₃ support without a noble metal under the present pretreatment conditions (40°C, 1 bar). Typically, the introduction of active

hydrogen onto the support requires H₂ treatment at much higher temperatures (above 300°C), which would inevitably affect the structural properties of the catalyst, making it difficult to conduct a blank experiment with the pure support. Additionally, the contribution of the support to activity can be further observed in selective poisoning experiments with CO (Fig. 3c).

Comment 6. Simplify the scope of the paper by removing the nitro-aromatic hydrogenations.

Response: Thanks for the reviewer's comment. Regarding the substrate scope presented in Figure 6, we aim to provide a universal strategy that is not limited to the semi-hydrogenation of alkynes but can also be applied to various other substrates. We believe that extending the range of applicable reaction types is more important than simply expanding the scope of substrates. While we have not extensively explored this aspect, our goal is to illustrate the universality of this strategy and, at the same time, inspire further research by offering a conceptual framework for designing catalysts for the selective hydrogenation of diverse substrates.

Comment 7. For Fig. 3g explain how, with a homogeneous crystallite distribution, some nanoparticles can be charged with H₂ to dissociate H₂ and trigger the spillover while some others are charged with alkynes.

Response: Regarding Figure 3g, the Pd nanoparticle in the center of the figure corresponds to the catalyst during the pre-treatment process, during which hydrogen dissociates on its surface and spills over to the support. This process forms a polar hydrophilic layer (HPL) composed of hydroxyl groups. The two sides of the figure actually represent a comparison of the catalytic performance of the catalyst before and after hydrogen pretreatment. On the right side, for the catalyst without pretreatment, the strong adsorption of the alkyne inhibits the dissociation activation of hydrogen. This behavior is also demonstrated by the reaction order experiments (Figure 3b). However, on the left side, for the pretreated catalyst, the hydroxyl groups on the support can participate in the hydrogenation of the alkyne, effectively preventing competitive

adsorption between the alkyne and hydrogen and thus weakening the poisoning effect of the substrate. We apologize for any confusion caused by the figure, and we have made some modifications to the figure accordingly.

Fig. R11. Schematic depiction of the contribution of overflowed H to hydrogenation through “hydrogen pool”. (This figure has been included in the revised manuscript as Fig. 3g.).

Comment 8. The TPDs (Fig 4d and 4e) show the effect of isolated atoms on adsorption equilibria. It would be interesting to compare with the Pd₁/MoWO₃ catalyst. A discussion citing the reference DOI: 10.1039/d1cy01016f which discusses in detail this topic would be a plus. Applying this methodology to the new catalysts studied in this article would be of premium interest.

Response: Following the reviewer’s suggestion, we conducted temperature programmed desorption (TPD) experiments for C₂H₂ and C₂H₄ on Pd₁/MoWO₃, as shown in Figure R12. It's clear that the adsorption of both alkene and alkyne is significantly reduced on the activated Pd₁/MoWO₃. This reduction can be attributed to the polar repulsion between the hydrophilic polar layer (HPL) and unsaturated molecules. It's worth noting that the adsorption of alkene on the active sites is much weaker than that of alkynes, which effectively prevents over-hydrogenation and improves the selectivity of the catalyst. We have included this figure in the supplementary information and discussed it in relation to the cited literature (Catal. Sci. Technol. 2019, 11, 6205-6216).

Fig. R12. Temperature-programmed desorption experiments of C_2H_2 and C_2H_4 over $Pd_1/MoWO_3$ and $Pd_1/MoWO_3-H$. (This figure have been added in the revised supplementary information as Supplementary Fig. 45)

Special changes are as follows:

Line 297-303 in the manuscript:

DFT calculation in Supplementary Fig. 44 confirm that the adsorption energy of MBE and MBY on both $Pd_1/MoWO_3$ and $Pd_1/MoWO_3-H$ significantly decreased. This reduction in adsorption energy is further proved by the temperature programmed desorption (TPD) of C_2H_2 and C_2H_4 (Supplementary Fig. 45). Notably, the adsorption of alkene is significantly decreased compared to alkyne after the formation of the HPL. This substantial reduction in alkene adsorption helps enhance the selectivity of the catalyst by preventing over hydrogenation^{23,64}.

Comment 9. 134: "The strong adsorption of alkyne can also result in self-poisoning of the catalyst" is new to me and not supported by any reference.

Response: Thanks for the reviewer's valuable comment. We apologize for the lack of clear description about the concept of "self-poisoning" and take delight to explain. In many instances, alkyne molecules exhibit strong binding to catalyst surfaces due to the unique nature of the unsaturated triple bond. This strong adsorption leads to a high coverage of alkyne molecules on the active sites of the catalyst (as discussed in references Appl. Catal. 1983, 6, 41–51, Appl. Catal. 1985, 15, 317–326, Nano Res. 2022, 15, 10044–10062, Catal. Sci. Technol., 2021, 11, 6205-6216).

Our calculations and the observed negative reaction order of MBY also support this phenomenon. Prior studies have shown that the adsorption energy of MBY on Pd (111) is higher than 1.50 eV (Catal. Sci. Technol., 2021, 11, 6205-6216, ACS Appl. Mater. Interfaces 2021, 13, 31775–31784, Angew. Chem. Int. Ed. 2022, e202202923). Excessive alkyne adsorption can lead to the blockage of active sites and can hinder hydrogen dissociation due to competitive adsorption, which can be described as a reactant-induced poisoning effect of the catalyst (Science 2017, 357, 389–393, ACS Catal. 2019, 10, 441–450, ACS Appl. Mater. Interfaces 2021, 13, 31775-31784, Appl. Catal. 1983, 6, 41–51).

We have rephrased the sentences and included references to further support this explanation.

Special changes are as follows:

Line 34-35 in the manuscript:

The strong adsorption of alkyne can also result in reactant-induced poisoning of the catalyst⁵⁻⁸.

Referee #3:

General Comment: Xiong et al. reported on a development of a catalyst for a selective hydrogenation of alkynes. A hydrophilic polar layer was induced via hydrogen spillover by choosing a reducible support for Pd atoms. The vertical effect of the polar layer guided the selectivity in the semihydrogenation of alkyne.

The development of a catalyst shows an elegant and novel design, that led to the targeted activity. The characterisation of the catalysts are through and comprehensive. The proposed mechanism is supported by the experimental data.

I believe that this manuscript deserves to be published in the Nature Communications journal.

Response: We would like to extend our heartfelt gratitude to the reviewer for their positive feedback and valuable insights.

REVIEWER COMMENTS

Reviewer #1 (Remarks to the Author):

All the issues requested by this Reviewer have been satisfactorily addressed. The article seems ready for publication. Just to comment that the cis/trans ratio in the new Fig. 6 is not clearly indicated.

Reviewer #2 (Remarks to the Author):

First of all, I would like to thank the authors for their answers to my previous comments.

* This article is interesting in that it confirms a known fact (<https://doi.org/10.1002/cctc.202300036>), i.e. a significant increase in alkene selectivity when hydrogenating alkynes with isolated palladium atoms. It also highlights a moderate acceleration of the reaction rate when palladium is doped with molybdenum.

* The best presented experimental yields and selectivity are comparable to others reported in the literature (DOI: 10.1039/C6RE00093B for example). But with these catalysts (Pd/WO₃) the reaction is, according to the authors, in negative order for concentrations below 0.1 M, which severely limits its activity at higher concentrations thus its practicability.

* The hypothesis of repulsive effects of the hydrophilic layer for alkenes is original but not supported by sufficient experimental evidence. Studying a catalyst undergoing activation is complex, and the whole demonstration is based on a single result. Many other effects could explain this phenomenon. The authors also talk about LSRs without providing any experimental justification. The authors also highlight the influence of spillover on this reaction, based on the different activity evolutions during the catalyst activation period, depending on the support. An alternative conclusion could be that supports that can be reduced consume hydrogen at the start of the reaction, resulting in a drop in activity for the hydrogenation of alkynes. What's more, once activated, the catalysts with different supports all have very similar activities (within a factor of 2), showing that this reaction is only weakly sensitive to support, which contradicts their hypothesis.

* It would be useful to have a summary table showing all the catalysts synthesized and their characteristics (in particular the effective amount of palladium). Indeed, much of the discussion is based on the amount of palladium for the different catalysts, with small variations, but the elemental analyses are not provided, so it would be important to check that differences in palladium loading do not explain certain results. This is particularly true of figure 3.e, where the regression of conversion as a function of palladium quantity does not pass through zero, which is astonishing. Moreover, the data for the different catalysts used in this figure are not provided.

Some other remarks :

- All experimental data should be provided in SI.
- Fig 3b. The authors' conclusion of a reaction order of -0.95 is biased by the simultaneous activation of the catalyst. There is an error in the axes of this figure: $\ln(c)$ are probably in mol/L and not mol/ml. What's more, the pure medium experiment (Fig. S.51) with a reaction rate of the same order of magnitude as that obtained at 0.1 M (implying an apparent order of 0) contradicts a negative order.
- Fig. 5d. and Fig. S46 Comparing selectivities without specifying at what conversion and concentration they were obtained is hazardous.
- Fig. S31b. the comparison between Sa-A and SA begs the question. The SA catalyst is fully activated after 60 minutes, so the rate of alkene hydrogenation should be the same as for SA-A. Yet it is more than double. However, it is more than double. This result casts serious doubt on the effect of activation on selectivity.

Miscellaneous remark : The authors use the unusual term "auto-inhibitory effect" to express that this reaction follows a Langmuir law with competitiveness between H₂ and acetophenone. This can be confused with "self-poisoning" reactions, where one of the reaction products is an inhibitor, which is not the case here.

[revised manuscript text omitted]

REVIEWERS' COMMENTS

Reviewer #2 (Remarks to the Author):

As the authors have elegantly demonstrated with figure R6 that hydrogenation of alkenes on their catalyst is inhibited by hydrogen, this work is complete and can be used as a basis for further research.

Nevertheless, the discussion of negative reaction orders during activation periods should really be toned down, as it is difficult to separate the two phenomena.

I thank the authors for their kind answers to my previous comments.

Point-by-Point Response to the Reviewer's Comments

We would like to express our gratitude to the reviewers for their insightful and thorough feedbacks of our manuscript (NCOMMS-23-32847B). Additionally, we extend our appreciation to the editor for handling our submission. We have diligently revised the manuscript in accordance with the reviewer's comments and suggestions, addressing each point individually. Below, we present our response to the referee's comments. We sincerely believe that the revised manuscript now meets the criteria for publication.

Detailed Response to Reviewer's Comments:

Referee #2:

General Comment:

As the authors have elegantly demonstrated with figure R6 that hydrogenation of alkenes on their catalyst is inhibited by hydrogen, this work is complete and can be used as a basis for further research.

Response:

We express our sincere gratitude to the reviewer for the positive comment.

Comment 1.

The discussion of negative reaction orders during activation periods should really be toned down, as it is difficult to separate the two phenomena.

Response:

Thank you for considering the suggestion. We acknowledge and accept your feedback, and we will limit the discussion of negative reaction order in the article. Our aim in discussing the change in reaction order during the activation process is to support the argument that the formation of HPL can effectively mitigate excessive adsorption of the alkyne, reduce poisoning effects, and consequently enhance reaction activity. It's worth noting that this conclusion is substantiated by additional experiments and computational results.